# WHAT MAKES LARGE LANGUAGE MODELS REASON IN (MULTI-TURN) CODE GENERATION?

**Kunhao Zheng**[1,2,*] **Juliette Decugis**[1,*] **Jonas Gehring**[1]**, Taco Cohen**[1],
**Benjamin Negrevergne**[2]**, Gabriel Synnaeve**[1]
[1]Meta AI (FAIR), [2]Paris Dauphine University - PSL
{kunhao, jdecugis, gab}@meta.com

## ABSTRACT

Prompting techniques such as chain-of-thought have established themselves as a popular vehicle for improving the outputs of large language models (LLMs). For code generation, however, their exact mechanics and efficacy are under-explored. We thus investigate the effects of a wide range of prompting strategies with a focus on automatic re-prompting over multiple turns and computational requirements. After systematically decomposing reasoning, instruction, and execution feedback prompts, we conduct an extensive grid search on the competitive programming benchmarks CodeContests and TACO for multiple LLM families and sizes (Llama 3.0 and 3.1, 8B, 70B, 405B, and GPT-4o). Our study reveals strategies that consistently improve performance across all models with small and large sampling budgets. We then show how finetuning with such an optimal configuration allows models to internalize the induced reasoning process and obtain improvements in performance and scalability for multi-turn code generation.

## 1 INTRODUCTION

The field of automatic code generation has made significant progress, particularly with the development of specialized Large Language Models (LLMs) (Chen et al., 2021; Li et al., 2022; Rozière et al., 2024; OpenAI, 2023; AI @ Meta, 2024). While these models have demonstrated proficiency in generating simple functions across various programming languages, there is still considerable room for improvement in their ability to tackle more complex algorithmic reasoning tasks, such as those found in competitive programming benchmarks like CodeContests (Li et al., 2022). Current state-of-the-art approaches either rely on model ensembling and massive single-turn sampling (Alpha-Code Team, 2023) or employ complex structured prompt chains for planning, editing and debugging (Ridnik et al., 2024; Islam et al., 2024). In contrast, multi-turn code generation strikes a balance between single-turn approaches and prompt chains, where code is built upon previous outputs in a dialog-like structure. This approach is motivated by applications such as LLM-based agents (Yao et al., 2023b), where models are tasked with decision-making and interacting with environments. In code generation, multi-turn approaches have primarily been explored on simple benchmarks or in small sample regimes due to their association with self-repair techniques (Olausson et al., 2024; Chen et al., 2024; Shinn et al., 2023; Zhong et al., 2024).

In this paper, we systematically deconstruct the components of previous research on prompting techniques and propose a unified framework for multi-turn code generation. Our objective is to establish a comprehensive and strong baseline, and to explore behavior and limitations across various sample regimes. Our focus on competition-level coding benchmarks and sample budgets is motivated as follows: (1) Popular methods such as chain of thought (Wei et al., 2022, CoT) yield improvements on reasoning-heavy tasks. However, they are designed to elicit reasoning traces for maximizing single-turn performance and are not inherently multi-turn. Competition-level benchmarks require algorithmic reasoning and thus provide an ideal testbed to evaluate whether CoT techniques can be extended beyond single-turn reasoning. (2) Recent studies suggest that the performance gains from self-repair are often modest when considering their generation cost (Olausson et al., 2024) and that repeated single-turn sampling serves as a strong baseline (Brown et al., 2024). As such, the trade-off

---

*Equal contribution.

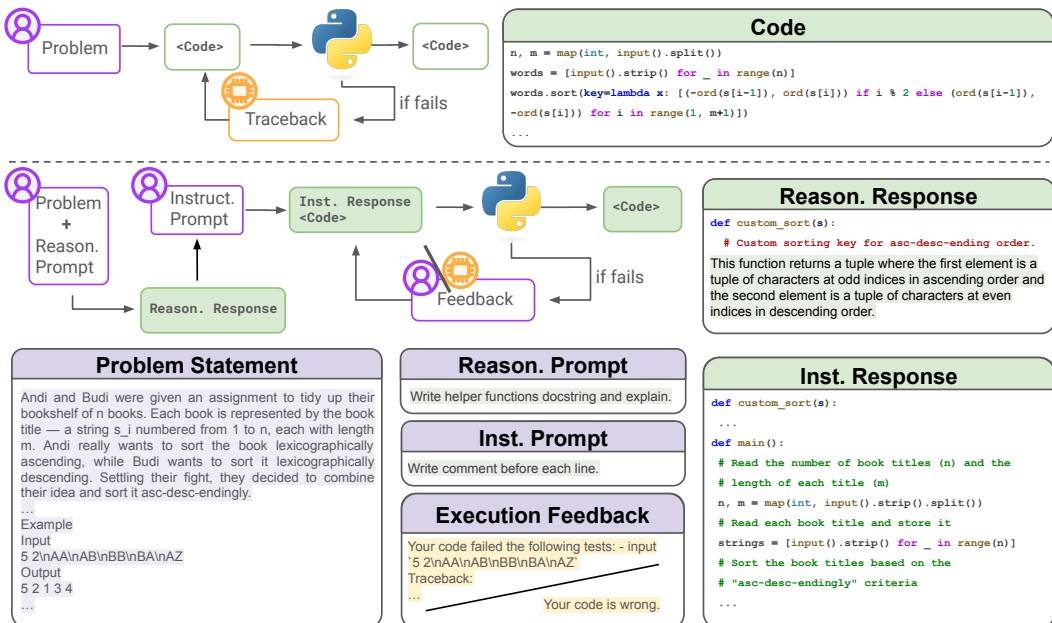

Figure 1: **Our framework for evaluating LLM multi-turn code generation techniques. Top**: In the default multi-turn setting, given a programming problem, the model generates a code solution, interacts with the runtime environment to gather execution feedback and retries in case of failure. **Bottom**: On top of the default setting, we gather *reasoning* (Reason.) prompts, *instruction* (Inst.) prompts, and *execution feedback* prompts. The problem statement is augmented with a *reasoning* prompt. After generating an answer to the *reasoning* prompt, an *instruction* prompt determines how program code should be generated. The *execution feedback* prompts vary in granularity, ranging from a binary pass or fail indicator to detailed tracing information.

between single-turn and multi-turn approaches, and the optimal allocation of resources between them, remains under-explored.

Our framework (Figure 1) enables mix-and-match combinations of single- and multi-turn code generation and chain-of-thought (CoT) techniques[1]: prompts that induce *reasoning*, such as a predicting problem attributes or writing natural language solutions first, and *instructions* that prompt different programming styles such as including comments or helper functions. Finally, we integrate *execution feedback* from intermediate solutions to allow for code repair. We conduct a comprehensive experimental survey across different benchmarks, LLM families and sizes, as well as sample regimes. Our analysis yields several key insights:

1. In the single-turn setting, combining *reasoning* prompts and *instruction* prompts achieves the best performance, and is more beneficial on larger models or harder problems. We also identify CoTs that degrade performance (Section 5.1).

2. The multi-turn setting alone brings modest gains and is sometimes worse than its single-turn counterpart under equal sampling budgets. The combination with CoT provides a significant performance boost on all models we study. Interestingly, detailed *execution feedback* prompts do not always translate to improved performance (Section 5.2). We show that this can be attributed to reduced diversity of generated programs which results in performance drops for large sample budgets.

3. LLMs can be instilled with reasoning behavior by finetuning on multi-turn CoT data (Section 5.3). The resulting model surpasses our best prompting configurations even without explicitly asking for CoTs during inference.

---

[1]We use the term "chain of thought" to refer to a broad family of prompting methods eliciting intermediate steps before or during code generation.

## 2 BACKGROUND

### 2.1 SINGLE-TURN VS. MULTI-TURN GENERATION: PROBLEM SETTING

We assume a coding problem $D = \{s, u, t\}$, where $s$ is the problem statement in natural language (e.g. see Figure 1), $u$ is a set of public tests, and $t$ is a set of private tests. A given code sample $c$ is considered correct if it passes *all* tests, or incorrect otherwise. Let $\pi$ denote an LLM that is able to produce a code sample $c$ for $D$ from a user prompt $p$ which includes the problem statement $s$. In the single-turn setting we thus obtain a code sample $c \sim \pi(\cdot \mid p)$.

In multi-turn code generation, we can distinguish between a *Natural-Language-to-Code* (NL $\rightarrow$ Code) task in the first turn and *Code-to-Code* (Code $\rightarrow$ Code) generation in subsequent turns. For a given problem, we generate a sequence of intermediary code samples $c_1, \ldots, c_T$ rather than just one. After each turn $i$, the code sample $c_i$ is fed back into the model $\pi$ together with an *execution feedback* prompt to obtain the next sample $c_{i+1}$. This process is repeated $T$ times until we either pass all public tests or until a maximum number of turns $N$ is reached. More formally, we can obtain every intermediary sample $c_i$, including the final code solution $c_T$, as follows:

$$c_i \sim \pi(\cdot \mid p_1, c_1, p_2, \ldots, c_{i-1}, p_i).$$

In this setting, the first prompt $p_1$ is the initial user prompt including the problem statement, and each $p_i$ for $i > 1$ is an *execution feedback* prompt containing the runtime result with error information or traceback optionally attached.

In the remainder of this study, the sequence $(p_1, c_1, ..., p_T, c_T)$ is denoted a *trajectory*, and the final code sample $c_T$ is called the *submission*. Only the code sample $c_T$ is tested against the private tests $t$ for correctness (i.e. intermediary code samples $c_i$ will only be tested against public tests $u$). Note that we sample not just one but several trajectories in parallel, starting with the same initial prompt $p_1$.

### 2.2 EVALUATION METRICS

We are interested in finding a correct solution to a given programming problem with a fixed budget, i.e., with a fixed number of code samples. For estimating the success rate of generated code samples, pass@$k$ is a widely used metric (Chen et al., 2021). For a problem $P$ and given a budget of $k$ samples, pass@$k$ is the expectation that at least one sample is correct, i.e., that it passes all tests.

**Limitations of pass@$k$** Pass@$k$ ignores computational requirements and thus puts single-turn evaluations at a disadvantage. In multi-turn settings, solutions are obtained via several generations (i.e., LLM calls) and hence at a higher cost, rendering these two setups not directly comparable (Kapoor et al., 2024).

In this study, we opt to measure performance via pass $n@k$ (Li et al., 2022) rather than pass@$k$ for a fair comparison of techniques. Pass $n@k$ estimates the success rate of a model $\pi$ on a problem $P$ using $k$ generations but at most $n$ submissions; it is the expectation that out of $n$ submissions one of them is correct (Appendix A). Following Li et al. (2022), we select $n$ submissions based on public test performance. Note that for $n = k$, both metrics are equivalent. For each benchmark, we report the average pass $n@k$ or pass@$k$ over all problems.

Figure 2 compares pass@$k$ and pass $n@k$ when measuring performance in a multi-turn setting. Pass@10 (**Top**) keeps increasing if we increase the maximum number of turns. How-

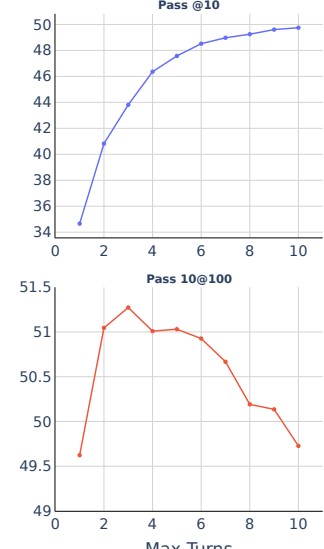

Figure 2: **Scaling number of turns is not compute optimal.** Pass@10 (**Top**) and pass 10@100 (**Bottom**) on CodeContests test set when increasing the number of turns with Llama 3.1 70B.

ever, pass 10@100 (**Bottom**) shows that compute optimality is lost after 3 turns. Given a budget of 100 samples with 10 programs selected as submissions, the optimal allocation of compute is obtained

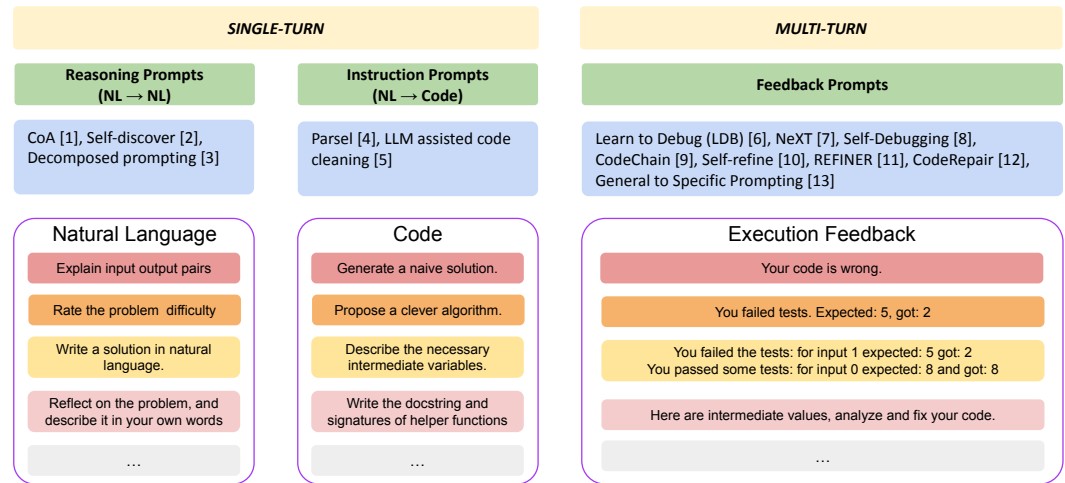

Figure 3: **Prompting space explored in our survey.** We explore chain of thought prompts at three different levels: before the first code generation (*reasoning* prompts), with code generation (*instruction* prompts), and after the first code generation (*execution feedback*). The corresponding works from the single-turn and multi-turn reasoning and code generation literature are: [1] Gao et al. (2024), [2] Zhou et al. (2024), [3] Khot et al. (2023), [4] Zelikman et al. (2023), [5] Jain et al. (2024b), [6] Zhong et al. (2024), [7] Ni et al. (2024), [8] Chen et al. (2024), [9] Le et al. (2024), [10] Madaan et al. (2024), [11] Paul et al. (2024), [12] Tang et al. (2024), [13] Li et al. (2023a).

by generating trajectories with 3 turns at most. As such, throughout this paper, we favor pass $n@k$ and report pass$@k$ only when comparing single-turn results exclusively.

## 3    PROMPTING AND FEEDBACK SPACE

We map the space of prompting techniques studied in our experimental survey in Figure 3. As CoT can intervene at different times in code generation, we categorize *reasoning* prompts (NL → NL) that elicit understanding of the problem before code generation, and *instruction* prompts (NL → Code) that guide the code output to enhance readability and modularity. These prompts can be applied in single-turn and multi-turn approaches.

In the multi-turn setting, we also introduce *execution feedback* prompts directly harvested from the runtime environment, serving as additional information for the model to self-repair within turns. We aim to determine the type of feedback which most effective on competitive programming benchmarks in the large sample regime. We thus evaluate several types of feedback, ranging in granularity:

- **Binary feedback**: A simple pass/fail indicator.

- **Failed tests feedback**: Provides expected and actual values for failed unit tests, along with tracebacks if any runtime errors are encountered.

- **Failed & passed tests feedback**: Expands on failed tests feedback by also including input/output information for passing tests.

- **LDB feedback** (Zhong et al., 2024): Offers debugger information, printing intermediate variable values and separating the code into blocks. The model must identify at which block the code failed and attempt to fix it.

CoT and execution feedback are incorporated into the generation through specific prompts as illustrated in Figure 1 (**Bottom**). As we will show in Section 5.2, different types of *execution feedback* induce different multi-turn behavior that can be classified as either *exploratory* or *exploitative*.

## 4 EXPERIMENTAL SETTING

**Models**   We perform experiments with the Llama Instruct series of LLMs, including Llama 3.0 and 3.1, 8B and 70B models (AI @ Meta, 2024). We use Llama 3.1 405B and GPT-4o in small sampling regimes only due to compute constraints.

**Single-turn**   Our grid search comprises 8 *reasoning* prompts and 6 *instruction* prompts, detailed in Appendix G. The *reasoning* prompts elicit intermediate steps either in natural language or with partial code. The *instruction* prompts either increase code readability ("describe"), break down the solution into modular code ("modularity"), or bias the type of solution ("solution"). Although we perform one more step of LLM inference for the *reasoning* prompts, we do not consider it an additional turn as our study compares the number of code attempts per problem and the effect of adding different types of extra tokens. We argue that this is equivalent to a single LLM call which groups all the *reasoning* prompts together, modulo the number of LLM forward passes. We generate with nucleus sampling (Holtzman et al., 2020, top-p=0.95) and a temperature of 1.0 to encourage output diversity.

**Multi-turn**   When performing multiple consecutive attempts at solving a coding problem, we set the code attempt limit to 3; this is motivated by the multi-turn results in Section 2.2, which reveal three turns as compute-optimal. We take the best *reasoning* prompts from the single-turn setting and combine them for up to 3 *reasoning steps* before code generation. We also introduce the *CoT-retry* setup, which allows for adaptive inference budget based on problem difficulty. In the first turn, we omit CoT prompts. If the first solution fails on more challenging problems, we prompt the LLM with a combination of *execution feedback* and a *reasoning* prompt. We employ a different prompt for each turn (see Appendix G.3). We also ablate different granularities of *execution feedback*. We do not include CoT prompts in this feedback comparison to isolate the effect of different feedback types.

**Rejection Sampling Finetuning**   With the Llama 3.1 70B model, we use the *CoT-retry* strategy to generate 3-turn trajectories on the CodeContests training set. We filter out trajectories with incorrect final code and perform supervised finetuning on the resulting data (details in Appendix B.2).

**Benchmarks**   We conduct our experiments on two competitive coding benchmarks in the zero-shot setting: (1) **CodeContests** (Li et al., 2022) contains 13k programming problems in the training set and 117/165 problems in the valid/test set. Each problem contains public tests, private tests, and generated tests. We use public tests to provide execution feedback in the multi-turn setting and use all available tests to evaluate the final submission. (2) **TACO** (Li et al., 2023b) is a collection of problems sourced from CodeContests, APPS (Hendrycks et al., 2021), and various programming contest platforms. The test set is split into 5 distinct difficulty levels: easy, medium, medium-hard, hard, and very-hard, with each level comprising 200 problems. This stratification allows us to examine the performance of different prompting strategies across difficulty levels. We use the first test case as the public test.

## 5 RESULTS

In this section, Table 1 and 2 first present maximum model performance for specific CoT variants. We then conduct a series of detailed experiments to better understand the performance impact of individual prompting methods. We structure our presentation by key findings outlined in Introduction.

### 5.1 SINGLE-TURN SETTING: CoT WORKS BEST FOR HARD PROBLEMS, LARGE MODELS, HIGH SAMPLING

We first investigate the impact of various CoT prompting strategies on models in the single-turn setting. There will be no *execution feedback* prompts. Therefore, our grid search involves searching in the space of *reasoning* prompts (NL → NL) and *instruction* prompts (NL → Code).

**Reasoning and instruction prompts can work together.**   We first compare the effect of various *reasoning* prompts, *instruction* prompts as well as combinations of both. Synthesized results are presented in Table 3, and we refer to Appendix C.1 for the complete set of experiments that led to Table 3. An interesting observation is that even the best performing *reasoning* and *instruction* prompts for pass@100 can decrease model performance in small sampling regimes (pass@1). Although *reasoning* prompts provide larger gains than *instruction* prompts (with the exception of Llama 3.1 70B), combining both results in the best performance.

Table 1: **Up to +10% pass $n@k$ with multi-turn CoT** on CodeContests test set with high temperature (1.0) and large sampling budget. In the multi-turn setting, we use a maximum of 3 code attempts (i.e., 3 turns) with the "failed tests" feedback. The pass $n@k$ is calculated from 200 trajectories for both single-turn and multi-turn settings. We also report the pass rates for Llama 3.1 70B after Rejection Sampling Fine-tuning (RFT) (Section 5.3). Prompts are the same across sample sizes per model.

| Model | Variants | CodeContests / Test | | | |
|---|---|---|---|---|---|
| | | 1@3 | 10@30 | 33@100 | 100@300 |
| Llama 3.0 8B | | 2.9 | 8.0 | 12.6 | - |
| | + CoT | $3.4_{+0.5}$ | $11.7_{+3.7}$ | $\mathbf{17.3_{+4.7}}$ | - |
| | + Multi-turn | $2.4_{-0.5}$ | $8.0_{+0.0}$ | $12.8_{+0.2}$ | 16.7 |
| | + Multi-turn CoT | $2.8_{-0.1}$ | $9.8_{+1.8}$ | $14.9_{+2.3}$ | 19.4 |
| Llama 3.0 70B | | 9.6 | 18.9 | 23.1 | - |
| | + CoT | $10.4_{+0.8}$ | $26.0_{+7.1}$ | $33.0_{+9.9}$ | - |
| | + Multi-turn | $10.1_{+0.5}$ | $21.0_{+2.1}$ | $26.7_{+3.6}$ | 32.7 |
| | + Multi-turn CoT | $11.1_{+1.5}$ | $26.5_{+7.6}$ | $\mathbf{34.3_{+11.2}}$ | 40.4 |
| Llama 3.1 8B | | 7.7 | 18.2 | 23.8 | - |
| | + CoT | $8.0_{+0.3}$ | $19.5_{+1.3}$ | $\mathbf{26.1_{+2.3}}$ | - |
| | + Multi-turn | $7.0_{-0.7}$ | $18.8_{+0.6}$ | $24.5_{+0.7}$ | 30.4 |
| | + Multi-turn CoT | $6.9_{-0.8}$ | $19.4_{+1.2}$ | $26.0_{+2.2}$ | 31.5 |
| Llama 3.1 70B | | 24.1 | 42.3 | 49.8 | - |
| | + CoT | $26.4_{+2.3}$ | $47.8_{+5.5}$ | $54.8_{+5.0}$ | - |
| | + Multi-turn | $24.1_{+0.0}$ | $43.8_{+1.5}$ | $51.6_{+1.8}$ | 56.2 |
| | + Multi-turn CoT | $27.7_{+3.6}$ | $48.4_{+6.1}$ | $\mathbf{55.3_{+5.5}}$ | 59.6 |
| Llama 3.1 70B[RFT] | | 26.2 | 45.1 | 50.9 | - |
| | + Multi-turn | $29.7_{+3.5}$ | $50.5_{+5.4}$ | $\mathbf{57.2_{+6.3}}$ | 61.1 |

Table 2: **Benchmarking of CoT across models: GPT-4o and Llama.** Pass 1@1 (%) and pass 1@3 (%) with low temperature (0.2). As models become more capable, repeated sampling surpasses a straightforward extension to multi turn (e.g. GPT-4o) or single-turn CoT (e.g. Llama 3.1 405B). A tailored multi-turn CoT, however, improves pass 1@3 performance across all models.

| Variants | GPT-4o | | Llama 3.1 70B | | Llama 3.1 405B | |
|---|---|---|---|---|---|---|
| | 1@1 | 1@3 | 1@1 | 1@3 | 1@1 | 1@3 |
| Single-turn | 17.0 | 27.6 | 23.2 | 27.3 | 27.8 | 32.9 |
| + CoT | $25.5_{+8.5}$ | $29.0_{+1.4}$ | $25.5_{+2.3}$ | $28.9_{+1.6}$ | $25.1_{-2.7}$ | $31.8_{-1.1}$ |
| + Multi-turn | - | $23.1_{-4.5}$ | - | $29.5_{+2.2}$ | - | $35.4_{+2.5}$ |
| + Multi-turn CoT | - | $\mathbf{31.5_{+3.9}}$ | - | $\mathbf{31.5_{+4.2}}$ | - | $\mathbf{40.1_{+7.2}}$ |

Table 3: **Combining *reasoning* and *instruction* works best** as compared to each individually for single-turn CodeContests test set (chosen based on pass@100 performance per model). In the best categories, results worse than the baseline are underlined.

| | Llama 3.0 8B | | Llama 3.0 70B | | Llama 3.1 8B | | Llama 3.1 70B | |
|---|---|---|---|---|---|---|---|---|
| | pass@1 | pass@100 | pass@1 | pass@100 | pass@1 | pass@100 | pass@1 | pass@100 |
| Baseline | 1.6 | 12.3 | 3.8 | 23.8 | 3.8 | 22.8 | 16.7 | 48.9 |
| Worst *reasoning* | 1.4 | 12.9 | 5.7 | 21.8 | 4.0 | 23.4 | 15.6 | 47.4 |
| Worst *instruction* | 1.4 | 11.3 | 3.4 | 25.1 | 3.7 | 20.9 | 14.9 | 48.4 |
| Worst Combination | 1.4 | 11.8 | 5.6 | 21.0 | 2.9 | 21.1 | 13.2 | 43.5 |
| Best *reasoning* | 1.8 | 15.7 | 7.0 | 30.4 | 4.1 | 25.7 | 15.7 | 52.2 |
| Best *instruction* | 1.3 | 13.5 | 5.5 | 29.6 | 3.6 | 24.6 | 16.8 | 53.8 |
| Best Combination | 1.5 | **17.3** | 5.3 | **33.1** | 4.0 | **26.1** | 16.1 | **54.1** |

**CoT is most helpful for large models.** With the smaller Llama 3.0 8B and Llama 3.1 8B, we observe from Table 3 that the best combination of *reasoning* and *instruction* prompts provides relatively small gains of 5.0% and 3.3% pass@100 on the CodeContests test set compared to the

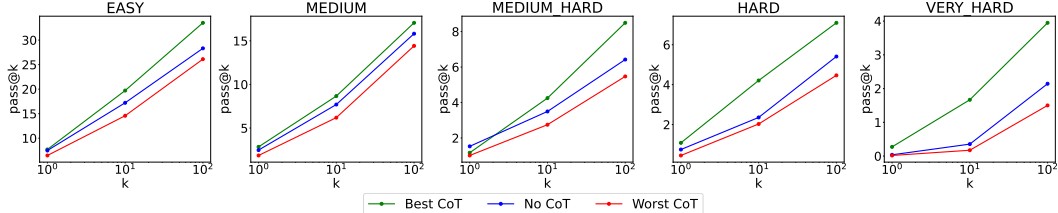

Figure 4: **CoT helps most on hard examples.** From a set of 8 reasoning and 6 instruction prompts commonly used on competitive coding benchmarks, we extract the pass rate of the best and worst prompts amongst all $63 = (8 + 1) \times (6 + 1)$ combinations (including no *reasoning* or no *instruction*) for Llama 3.0 8B. We compare on different difficulty split of the TACO dataset. The relative gain from a tailored CoT increases with problem difficulty and sampling size.

improvements of 9.3% and 5.2% from the corresponding 70B models. Interestingly, we found that not all sets of prompts are beneficial. the worst combination degrades the pass@100 of Llama 3.1 70B by up to 5.4%. CoT makes performance worse if the model fails to follow the *instructions* or makes the LLM propose a sub-optimal plan. Sub-optimal plans are usually brute force approaches to solve the problem which do not fit the time limits constraint (see Appendix H for an example).

**CoT is most helpful for harder problems.** With the TACO dataset, which provides a difficulty split, we can observe that CoT does help smaller models on harder problems. Figure 4 demonstrates that the relative gain from the best *reasoning* and *instruction* prompt combination, compared with the baseline performance (No CoT), increases with problem difficulty. For example, the pass@100 of Llama 3.0 8B nearly doubles with CoT on the very-hard test split (2.1% → 3.9%). We show in Appendix C.3 that this observation generalizes to Llama 3.1 8B and 70B model.

**Prompt efficacy is model and sample size dependent.** No singular *reasoning* and *instruction* combinations work best across sampling sizes and models (see Appendix C.2 for detailed analysis). *Reasoning* prompts that simplify the problem (e.g., self-reflection, explain input-output pairs) benefit smaller models (8B models) whereas larger models (70B, 405B, GPT-4o) gain most from generating parts of the solution (e.g., write function docstrings). "Solution"-based *instruction* prompts are the most efficient across models, specifically for the Llama 3.1 series, as shown in Figure 5.

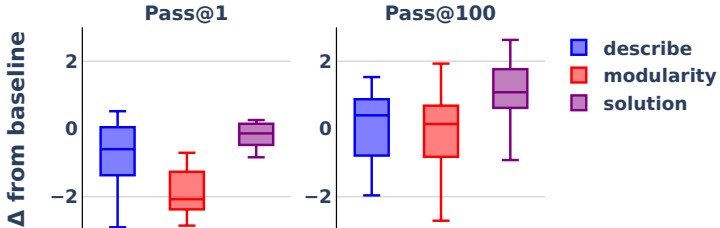

Figure 5: **Solution-based *instruction* prompts work best across Llama 3.1 models.** We separate *instruction* prompts into "describe" (e.g., add comments, imports), "modularity" (e.g., add helper functions) and "solution"(e.g., write a naive solution, propose a clever algorithm). The performance difference ($\Delta$) is normalized with respect to the baseline and standard deviation per pass rate.

## 5.2 MULTI-TURN SETTING: SELF-REPAIR LIMITED WITHOUT COT AND PROPER FEEDBACK

We summarize our multi-turn results in Table 1. With a fixed number of samples, i.e., $k$ in pass $n@k$, multi-turn alone provides modest gains only (usually less than +2%) and sometimes even reduces pass $1@3$ performance compared to drawing independent samples in single-turn mode. Notably, this is the case for smaller models (Llama 3.0 and 3.1 8B). In this section, we take a closer look at performance drops in the multi-turn setting and explore methods that can take advantage of accessing previous wrong solutions.

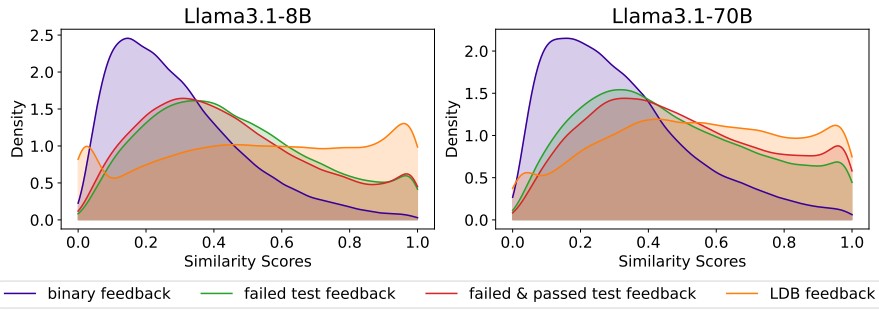

Figure 6: **Fine-grained feedback induces exploitative behavior.** Distribution of consecutive code similarity scores within dialog for different types of feedback, obtained from Llama 3.1 8B and 70B samples (temperature 1.0). The higher the similarity scores between consecutive codes in the same dialog, the more the model exhibits exploitative behavior.

**Reasoning prompts are not additive.** It is tempting to consider that stacking more *reasoning* prompts before code generation will further guide the model towards correct solutions. For example, prompts might increase the granularity of reasoning: self-reflect on the problem, explain the input/output pairs, write helper functions, and finally output a full code solution. However, we empirically find that across models, one step of *reasoning* provides the most significant boost. The performance plateaus or even decreases with two or three steps. Increasing the number of *reasoning* steps hurts both Llama 3.0 and 3.1 models (see Table 7 in Appendix D.1). For the best models, a single step with a *reasoning* prompt is most beneficial.

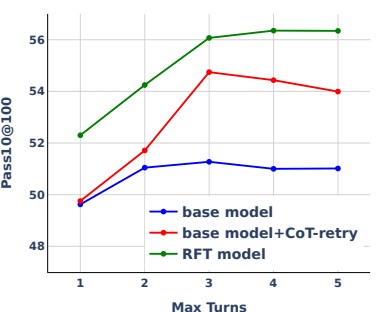

Figure 7: *Reasoning* and *execution feedback* prompts, and RFT, enhance both single- and multi-turn performance for Llama 3.1 70B.

***CoT-retry* works best.** For Llama 3.0 models, simply extending the single turn *reasoning* and *instruction* prompts to the multi-turn setting yields superior performance (reported as "Multi-turn CoT" in Table 1). However, as models become more capable, an increasing number of problems in CodeContests are solved in the first attempt without specific prompts. *CoT-retry* only *reasons* when the first attempt fails and therefore works best across Llama 3.1 models for all sampling sizes and benchmarks ("Multi-turn CoT" in Table 1). Figure 7 decomposes its per-turn performance. When extending the number of turns from 2 to 3, Llama 3.1 70B alone shows diminishing gain while combination with *CoT-retry* still increases the performance by a large margin.

**Execution feedback granularity determines exploration-exploitation behavior.** Given previous incorrect code and execution feedback, subsequent attempts can consist of a fresh attempt (exploration) or of updates to prior solutions based on feedback (exploitation). We quantify this behavior by computing similarity scores between two consecutive solutions (details in Appendix B.1). Figure 6 shows that with more fine-grained information provided via execution feedback, models exhibit exploitative behavior (high similarity scores). Exploitation can be a desired property on relatively easy problems where errors are due to simple bugs. However, we posit that diversity is key to improving performance on difficult problems, i.e., exploratory behavior within a trajectory based on the *execution feedback* prompts. This matches our experimental results: simple execution feedback (e.g., binary, failed tests) provides optimal performance for most models (Appendix D.2).

### 5.3 COT REJECTION SAMPLING FINE-TUNING: MODELS CAN INTERNALIZE REASONING

We investigate whether LLMs can benefit from finetuning on reasoning traces obtained via CoT prompting. We thus perform Rejection Sampling Finetuning (RFT) on Llama 3.1 70B, where the *reasoning*, *instruction* and *execution feedback* prompting strategies we consider act as *policy improvement operators*: they elicit the model's reasoning ability and produce a higher number of trajectories

Table 4: **Multi-turn CoT and RFT generalize to TACO test set.** Pass $n@k$ (%) of Llama 3.1 70B on multi-turn TACO test set with temperature 1.0. We use the best multi-turn CoT found on CodeContests. We use the model RFTed on CodeContests training set (after decontamination, details in Appendix I) and report its performance directly on TACO without CoT.

| Model | easy | | medium | | medium_hard | | hard | | very_hard | |
|---|---|---|---|---|---|---|---|---|---|---|
| | 1@3 | 100@300 | 1@3 | 100@300 | 1@3 | 100@300 | 1@3 | 100@300 | 1@3 | 100@300 |
| Llama 3.1 70B | 31.6 | **60.2** | 14.2 | 44.6 | 9.5 | 36.2 | 4.4 | 20.6 | 1.8 | 9.0 |
| + Multi-turn CoT | 32.3 | 59.8 | 15.0 | **46.2** | 10.8 | 38.5 | 5.8 | 22.8 | 2.6 | 11.8 |
| Llama 3.1 70B$^{RFT}$ | **34.1** | 58.9 | **18.0** | 45.3 | **13.0** | **39.4** | **8.1** | **23.3** | **3.5** | **12.0** |

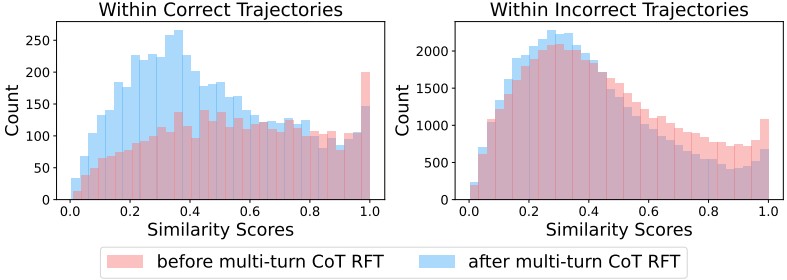

Figure 8: **RFT makes the model produce more diverse code within trajectories** as shown by the consecutive codes' similarity scores before/after RFT on CodeContests test set evaluated with multi-turn no CoT. This shift towards more exploratory behavior contributes majorly to the gain of correct trajectories.

with correct submissions. Given the low variance across different feedback types (Table 8 in Appendix D.2), we opt for simplicity and use the "failed tests" execution feedback combined with *CoT-retry* for data generation.

More specifically, we improve a model $\pi$ by 1) collecting a dataset of correct trajectories sampled from $\pi$ with CoT enabled at inference time, 2) removing the CoT prompt in the collected trajectories, and 3) finetuning $\pi$ with the standard next-token prediction objective. With this strategy, we can now obtain CoT-level trajectories without adding specific prompts at inference time.

Figure 9, Table 1, and Table 4 show that the RFT model provides additional gains over inference methods across sampling sizes and datasets. Beyond performance, RFT on multi-turn CoT improves sampling diversity (Figure 8) and self-repair capacities, especially for long trajectories (Figure 7). Behavior-wise, we show in Table 11 (Appendix F.1) that RFT results in model responses with increased textual content.

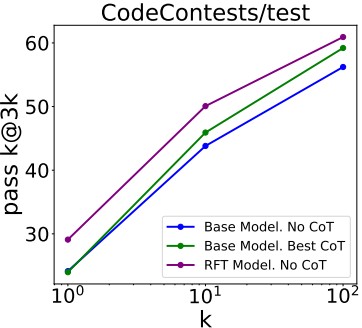

Figure 9: Llama 3.1 70B's pass $k@3k$ on CodeContests. *CoT-retry* increases the performance in large sampling regimes. RFT transfers this reasoning ability to no CoT setting and lifts the pass rate curve across sampling budgets.

## 6 RELATED WORK

**Chain of Thought with Code**    Chain of Thought (CoT) enables step-by-step thinking for LLMs to solve mathematical word problems in either few-shot (Wei et al., 2022) or zero-shot (Kojima et al., 2022) settings. Many variants, e.g., Tree of Thought (Yao et al., 2023a), have emerged in code generation since. Chen et al. (2023b) and Gao et al. (2023) translate natural language mathematical problems in executable code for the model to separate reasoning and computation. These methods rely on the LLM outputting correct code to represent a problem. We see this work as tangential to ours as boosting LLM coding performance will also help on overall reasoning tasks. Higher levels of abstractions (Khot et al., 2023; Zhou et al., 2024; 2023; Zelikman et al., 2023; Jain et al., 2024b) and

self-repair techniques (Paul et al., 2024; Li et al., 2023a; Ridnik et al., 2024) have been proposed. Beyond inference methods,Wadhwa et al. (2024); Yu et al. (2024); Zelikman et al. (2022); Hosseini et al. (2024); Pang et al. (2024) explore new training algorithms and loss functions to learn from CoT. In comparison, we bring novelty to the type of CoT used in training (multi-turn) and rely on simple Rejection Sampling Fine-tuning (RFT) (Touvron et al., 2023; Yuan et al., 2023; AI @ Meta, 2024). It has been shown to achieve good performance, with less data compared to SFT (Setlur et al., 2024).

**Execution feedback**    Currently LLMs struggle to understand code execution feedback (Gu et al., 2024) as this type of data is rarely present in their training set. Zhong et al. (2024) and Ni et al. (2024) try to mimic "print debugging" to convey intermediate code steps to the LLM. Olausson et al. (2024) found that the effect of self-repair largely depends on the text quality of the subsequent reasoning and therefore use only textual feedback. In our setting, we are interested in the feedback which could be directly harvested from the execution environment. Shi et al. (2022); Li et al. (2022); Chen et al. (2023a) likewise proposed unit test generation as a way to increase coverage with execution feedback. Adding test generation to our pipeline would be an interesting avenue for further work.

**Inference Optimization**    With the rise of LLM agents (Kapoor et al., 2024) and the scaling effect of test time techniques (Li et al., 2022; Snell et al., 2024; Brown et al., 2024), inference optimization against compute resources becomes increasingly relevant. Similar to our pass $n@k$ argument in Section 2.2, Kapoor et al. (2024) discuss the importance of controlling for generation cost in AI agent evaluations.

## 7 LIMITATIONS

In our multi-turn setting, we do not explore further branching at the second or third turn, i.e., more complex tree structures (Tang et al., 2024) or in general inference-based search approaches (Snell et al., 2024), e.g., with look-ahead or backtracking, as we focus on the effect of additional CoT tokens generation. Although a maximally fair comparison (at the cost of complexity) should account for total input and output tokens (Olausson et al., 2024) as well as model size (Hassid et al., 2024), we believe pass $n@k$, which stresses the number of code attempts, constitutes a simple yet superior alternative to pass@$k$. Our RFT is similar to Expert Iteration (Anthony et al., 2017) and ReST (Gulcehre et al., 2023) when considering a single iteration only. We also assume trajectories with correct final code contain correct reasoning. Adding a Process-Reward Model (PRM) or a "critic" LLM (Zheng et al., 2024) to rate and filter the correctness of the reasoning tokens could enhance training data quality and diversity. Future work could benefit from exploring more advanced inference techniques such as prompt tuning (Lester et al., 2021) or training strategies such as including "near-correct" trajectories (Pang et al., 2024; Setlur et al., 2024) with multi-turn CoT. Finally, we speculate that the effectiveness of different prompts for different LLM families (particularly the Llama 3.0 vs. 3.1 series vs. GPT-4o) could be attributed to the mixture of finetuning data (Chung et al., 2022). Exploration of this topic is beyond the scope of this paper.

## 8 CONCLUSION

In this work, we present a comprehensive experimental survey on various *reasoning*, *instruction* and *execution feedback* prompts in the single-turn and multi-turn code generation task at scale. Our results on two competitive programming benchmarks, CodeContests and TACO, suggest that incorporating CoT techniques, originally designed for single turns, and *execution feedback* prompts into the multi-turn setting is non-trivial. Due to the difficulty of the benchmarks, a major contributor to performance is problem understanding rather than the ability to perform code repair with detailed feedback. With a set compute budget, using multiple turns alone can hamper performance compared to repeated sampling with high temperatures. Biasing the model with adapted CoT based on problem difficulty at each turn boosts its self-repair abilities and leads to consistent gains across all model series and sizes. Beyond inference methods, our RFT experiment shows that multi-turn reasoning traces triggered by prompts can be internalized, which leads to advanced reasoning abilities. We hope that our findings motivate further research in more advanced multi-turn settings. One example is repository-level code agents, where models interact with complex environments to gather feedback and extensive planning and reasoning capabilities are demanded.

REPRODUCIBILITY STATEMENT

As our paper focuses on inference methods with existing models, the key components for reproducibility are access to models, datasets, and prompt descriptions. All the models (except our fine-tuned RFT model) used in this paper are publicly available at the time of writing: Meta Llama 3.0 and 3.1 series are open-weight, and gpt-4o-2024-05-13 (GPT-4o in the paper) are available through OpenAI API. The two benchmarks we use: CodeContests (`https://github.com/google-deepmind/code_contests`) and TACO (`https://github.com/FlagOpen/TACO`) are publicly available. We provide a complete list of all our prompts in Appendix G to reproduce single-turn and multi-turn experiments. We present the details of computing similarity score with normalization in Appendix B.1. Regarding finetuning, our main contribution relies on the data augmentation technique on Code-Contests for which we present the details in data collection, deduplication, and decontamination approach, as well as statistics such as the number of trajectories and the number of total tokens in Appendix B.2, B.3 and I. We detail our finetuning hyperparameters in Appendix B.2 to reproduce our RFT model training. We will release the code for our multi-turn and CoT methods to facilitate reproduction.

ACKNOWLEDGEMENT

We thank Quentin Carbonneaux, Baptiste Rozière, Jade Copet, Olivier Duchenne, Fabian Glöeckle, Badr Youbi Idrissi, Nicolas Usunier, Sten Sootla, Chris Cummins, Sida Wang, Pierre Chambon, Matthieu Dinot, Ori Yoran, Kush Jain, Naman Jain and all the members in FAIR CodeGen team for helpful technical contributions, suggestions, and insightful discussions. We thank the Infra team for the support for enabling a seamless compute cluster experience.

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

## A   FORMULA AND ALGORITHM FOR PASS $n@k$ METRICS

Formally, let $N$ be the total number of code samples. Let $F$ be the number of codes filtered by public tests, among which there could be false positives. Let $C$ be the number of correct codes that pass all the unit tests. The pass $n@k$ for a benchmark of problems is defined as follows:

$$
\text{pass } n@k = \mathbb{E}_{\text{Problems}} \left[ 1 - \sum_{i=0}^{k} \left( \frac{\binom{F}{i}\binom{N-F}{k-i}}{\binom{N}{k}} \right) \left( \frac{\binom{F-C}{n_p}}{\binom{F}{n_p}} \right) \right],
\tag{1}
$$

where $n_p = \min(i,\ n)$.

**Explanation**   The first term $\frac{\binom{F}{i}\binom{N-F}{k-i}}{\binom{N}{k}}$ is the probability of having $i$ filtered solutions among $k$ solutions, which obeys a hyper-geometric distribution, $\text{HYPERGEOMETRIC}(F,\ N-F,\ k)$. Given the number of submissions $n_p = \min(i,\ n)$, the second term $\frac{\binom{F-C}{n_p}}{\binom{F}{n_p}}$ is the probability of having none of the correct solutions.

In evaluation, instead of computing the combinatorial number, we use Monte Carlo estimation by re-sampling $k$ solutions $n_{\text{boot}}$ times for bootstrapping (in our case, we use 10000). The algorithm for such is described in detail in Appendix A.3 of the Alphacode paper (Li et al., 2022).

## B   REJECTION FINE-TUNING EXPERIMENT DETAILS

### B.1   COMPUTING SIMILARITY SCORE

We compute the similarity score of two Python code snippets as follows.

First, we pre-process the code snippet to remove formatting and variable naming effects. We normalize variable names by running an in-order indexing scheme on the Abstract-Syntax-Tree (AST), as shown in Figure 10, followed by simple formatting by `lambda x: ast.unparse(ast.parse(x))`. We note that there are 1%-2% of codes failing the parsing because of syntax error, in which case we skip this normalization step.

```
total_n = int(input())                                  v_0 = int(input())
for _ in range(total_n):                 ⟶              for v_1 in range(v_0):
    S_num, P_num = list(map(int, input().split()))          (v_2, v_3) = list(map(int, input().split()))
```

Figure 10: Example of variable renaming AST pass.

Second, we use `difflib.SequenceMatcher` to compute the similarity score for the normalized snippets.

### B.2   RFT DATA COLLECTION

Our data collection pipeline consists of 3 major steps: generation, filtering and post-processing, deduplication and decontamination. We present the details of each step, including the parameters we use and the dataset statistics.

### B.2.1   GENERATION

Using *CoT-retry*, we generate 200 multi-turn trajectories with a maximum of 3 code attempts using Llama 3.1 70B for each problem instance in CodeContests training set. The generation is in the standard chat format for Llama 3.1 series[2]. We do not include the system prompt in the dialog. We use nucleus sampling (Holtzman et al., 2020) with top-P=0.95 and temeprature 1.0.

---

[2] https://www.llama.com/docs/model-cards-and-prompt-formats/llama3_1/

We follow the same format as the evaluation: Final code solutions are tested against all the tests, and the code solutions in the middle of the dialogs are tested against public tests. If the model solves the problem in the first turn, the trajectory will still be collected while there will not be *execution feedback*.

### B.2.2 FILTERING AND POST-PROCESSING

After filtering the incorrect trajectories, we keep only $60\%$ of all the generated trajectories where the code in the last turn passes all the tests. We assume that correct final code correlates with correct reasoning in the CoT and self-repair techniques. The set of successful trajectories contains solutions to 7238 problems in the CodeContests training set (in total 13213 problems), among which 1105 problems are only solved under the multi-turn setting. Interestingly, we found 485 problems which could be solely solved under the single-turn setting of all the generated 200 code trajectories.

We apply additional post-processing to the trajectories by removing the CoT prompt introduced but keep the model response untouched. This enables the model to develop inherent CoT-like reasoning capabilities through fine-tuning.

We separate the successful trajectories into 2 sets: single-turn trajectories and multi-turn trajectories. The single-turn trajectories contain 426952 trajectories, solutions to 6133 problems. The multi-turn trajectories contain 226382 trajectories, solutions to 6753 problems.

### B.2.3 DEDUPLICATION AND DECONTAMINATION

We conduct LSH-based deduplication on each set to the code solutions per problem instance to a maximum of 50 solutions, by following the practice of Jain et al. (2024b). We use hash size 64, jaccard threshold 0.5, number of bands 60 and band size 5 for the LSH-based deduplication configuration.

We further conduct a decontamination between the collected solutions and TACO test set (details in Appendix I). This enables a direct evaluation of the finetuned model on TACO test set to measure the generalization to TACO.

After deduplication and decontamination, we harvest 177475 single-turn trajectories (in total 143M tokens) and 160600 multi-turn trajectories (in total 285M tokens).

### B.3 FINETUNING SETTING

We perform self-supervised fine-tuning on the above-mentioned multi-turn trajectories using Llama 3.1 70B. We use standard cross-entropy loss on the last full body of the model response in the last turn and treat all the previous user and model messages as the prompt part.

The finetuning uses learning rate $2\mathrm{e}^{-6}$, 545 steps of gradient updates, sequence length 8192, global batch size 524288 tokens. We use AdamW as the optimizer with weight decay 0.1, $\beta_1 = 0.9$ and $\beta_2 = 0.95$. The learning rate schedule is cosine scheduling with 10 warmup steps annealing to 10% of peak learning rate at the end of the training. We do not do early stopping to evaluate the model checkpoint in the middle of the finetuning. Instead, our evaluation always compares model checkpoints under different configurations at the end of the funetuning.

The end-to-end finetuning takes 170 H100 · hours with Tensor Parallelism of size 8 and Fully Sharded Data Parallelism (FSDP).

### B.4 GENERALIZATION OF RFT MODEL

Beyond competitive programming tasks such as CodeContests and TACO, we studied whether our RFT model, Llama 3.1 70B$^{\text{RFT}}$, fine-tuned on CoT and multi-turn data generalizes to other code generation tasks. Table 5 and Table 6 show results for the single-turn and multi-turn experiments, respectively. For single turn, we report performance on the following code generation benchmarks: HumanEval+ (Chen et al., 2021; Liu et al., 2024), MBPP+ (Austin et al., 2021; Liu et al., 2024) and LiveCodeBench-v4 (Jain et al., 2024a). We also report multi-turn performance on LiveCodeBench-v4. Our RFT model performs similarly, sometimes with slight performance degradation, and often better

than Llama 3.1 70B, which shows that the model does not overfit to CodeContests and generalizes to unseen code generation benchmarks.

Table 5: **RFT model fine-tuned on CodeContests generalizes to other code generation datasets.** Each line corresponds to single-turn performance evaluated without CoT prompts for both models. Results are reported under the format pass@1 / pass@10. We use temperature 0.2 for sampling.

| Model | HumanEval+ | MBPP+ | LiveCodeBench - v4 | | | |
|---|---|---|---|---|---|---|
| | | | Easy | Medium | Hard | All |
| Llama 3.1 70B | 71.8 / **77.0** | 65.2 / **70.9** | 73.8 / 85.0 | 22.0 / **37.4** | 3.3 / 7.2 | 34.2 / 45.3 |
| Llama 3.1 70B$^{RFT}$ | **72.1** / 76.9 | **63.5** / 69.2 | **76.2** / **85.7** | 22.0 / 37.0 | **3.5** / **8.0** | **35.1** / 45.3 |

Table 6: **Better low sampling multi-turn performance with the RFT model.** We prompt models without CoT and perform multi-turns with a maximum of 3 turns. Results are reported under the format pass 1@3 / pass 10@30. We use temperature 0.2 for sampling.

| Model | LiveCodeBench - v4 | | |
|---|---|---|---|
| | Easy | Medium | Hard |
| Llama 3.1 70B | 82.8 / 94.3 | 30.8 / 49.2 | **4.77** / **9.45** |
| Llama 3.1 70B$^{RFT}$ | **86.0** / **94.4** | **31.5** / **50.1** | 4.74 / 9.19 |

## C  ADDITIONAL SINGLE-TURN RESULTS

### C.1  GRID SEARCH RESULTS

We provide the complete grid search results for all our *reasoning* and *instruction* prompts across all models and pass rates for the single turn setting. This demonstrates the variability in effectiveness per sampling size and LLM series. The "weak solution" *instruction* prompt is a clear winner for larger sampling sizes $k \geq 10$. We show in Figure 11, 12, 13 and 14 the grid search of all *reasoning* and *instruction* prompts for the Llama 3.0 and 3.1 series. As we increase the sampling budget, we increase the sample diversity and the recall across all CoT. For a low sampling budget, most prompts hurt performance. CoT is the most effective with Llama 3.0 70B.

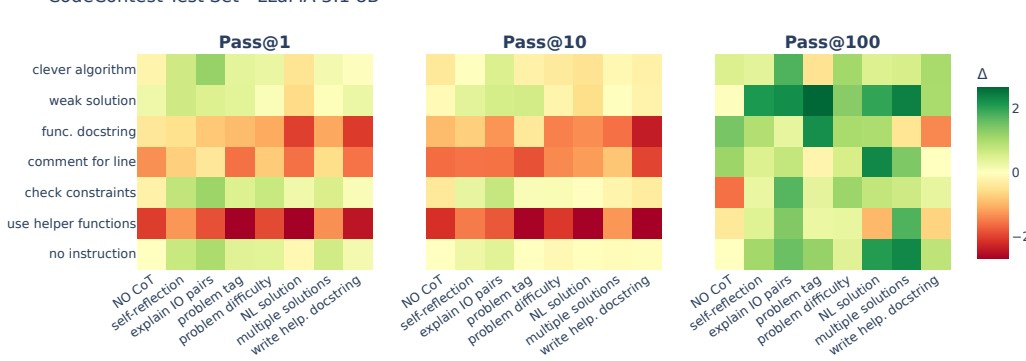

Figure 11: Grid search of all *reasoning* and *instruction* prompts for Llama 3.1 8B.

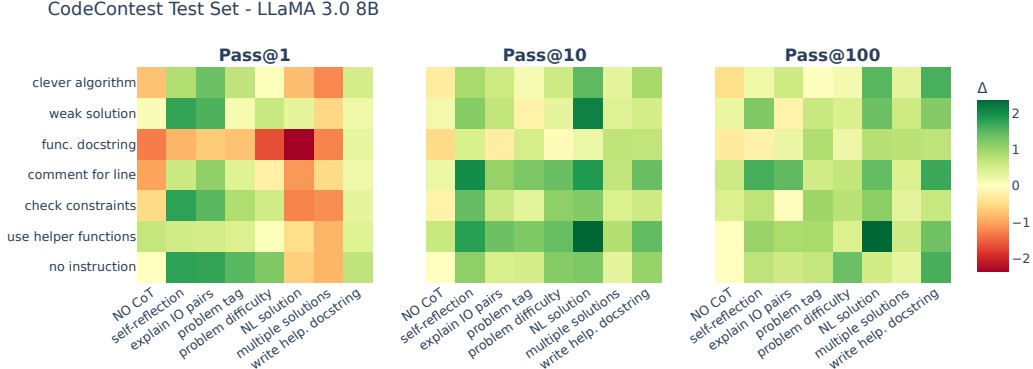

Figure 12: Grid search of all *reasoning* and *instruction* prompts for Llama 3.0 8B.

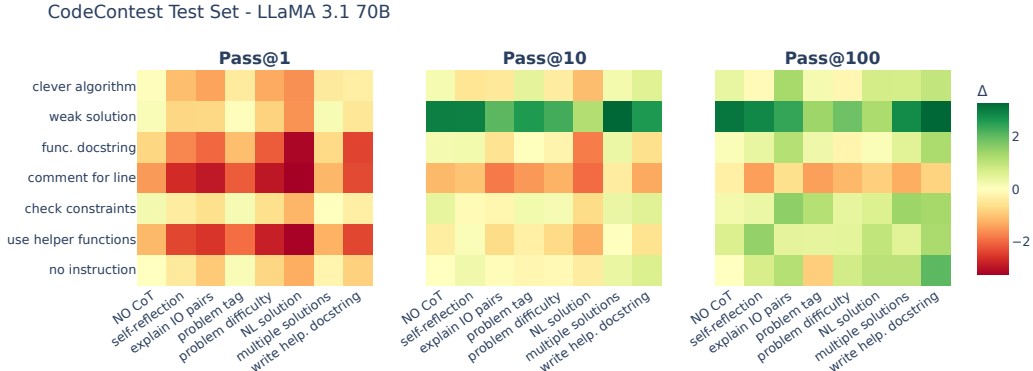

Figure 13: Grid search of all *reasoning* and *instruction* prompts for Llama 3.1 70B.

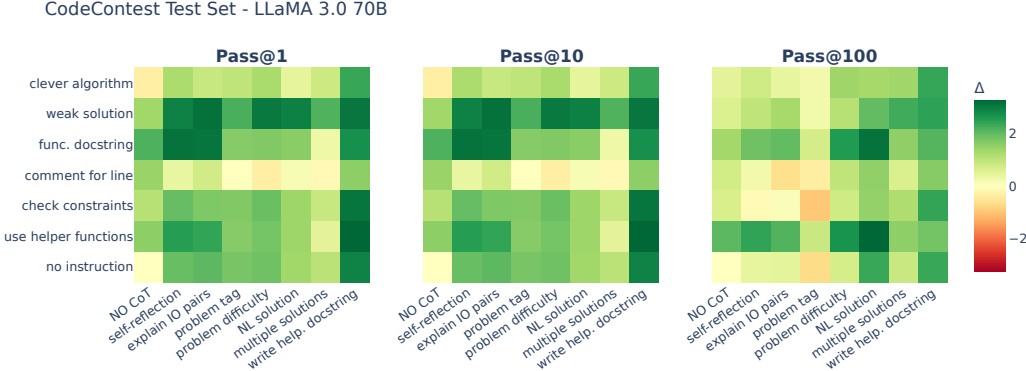

Figure 14: Grid search of all *reasoning* and *instruction* prompts for Llama 3.0 70B.

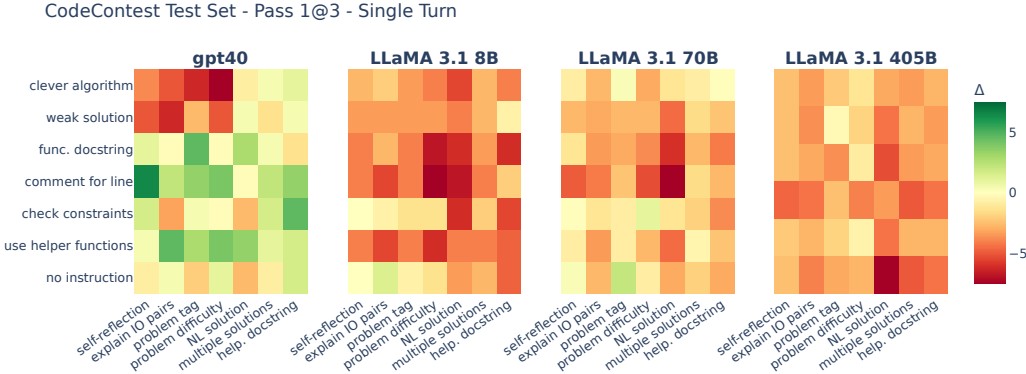

Figure 15: **No gold CoT across models.** Based on our grid search of instruction and reasoning prompts, we compare all 63 single-turn results across three different models. With a low sampling budget, most prompts perform similarly, if not worse than the baseline performance (without CoT). The best prompt (in green) differs for each model, but we see similar patterns in the Llama models.

## C.2 DETAILED ANALYSIS OF SINGLE-TURN PROMPTS

When comparing *reasoning* and *instruction* prompts, the values are normalized with respect to the baseline in each respective pass rate specifically: $x \leftarrow \frac{x - \text{baseline}}{\text{std}(\mathbf{x})}$. The value at $0$, therefore, corresponds to no *reasoning* and no *instruction* prompts. We provide further results aggregated across models and types of prompts.

As demonstrated by Figure 16 and Figure 17, we have large variations across models and prompt types and observe that no *reasoning* and *instruction* prompt always performs above the 0 baseline. As shown in Figure 18, the best combinations often rely on "weak solution" *instruction* but vary across sample sizes for *reasoning* with "self-reflection" for lower sampling budget and "helper functions" for higher sampling budget. We observed writing intermediate variables before code often made performance worse and could be qualified as the "worst" *reasoning* prompt for all models.

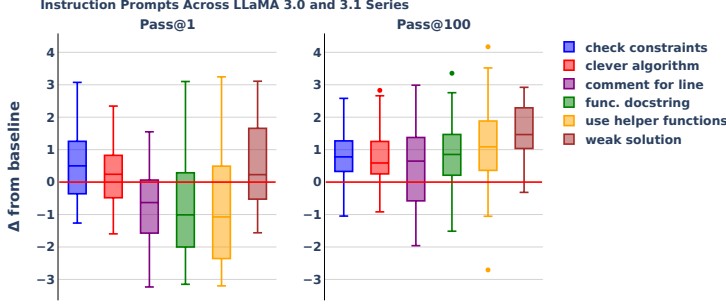

Figure 16: **Group by *instruction* prompts** averaged across all *reasoning* prompts for the Llama 3.0 and 3.1 models. We observe that "check constraints" is a winner for pass@1 and "weak solution" for pass@100. Overall, "add a comment before each line" seems the least efficient across models.

## C.3 GENERALIZATION OF SINGLE-TURN BEST COT TO LLAMA3.1 MODELS

We show in Figure 19 that the best CoT (i.e., *reasoning* and *instruction* prompt and their combination) found with Llama 3.0 8B on TACO could be directly ported to Llama 3.1 8B and 70B models. We also observe that CoT brings more boost on harder problems by comparing the relative gain of pass rate on the easy and very-hard split.

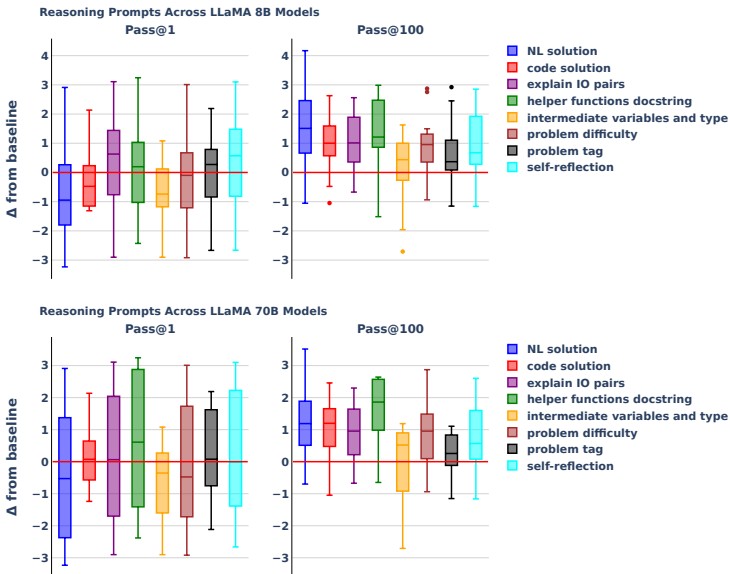

Figure 17: **Group by *reasoning* prompts** averaged across all *instruction* prompts (top) for small models and (bottom) for large models. For pass@1, "explain IO pairs" helps small models, and "helper function docstrings" helps large ones. The relative efficacy of each prompt converges to a similar order for pass@100 for large and small models.

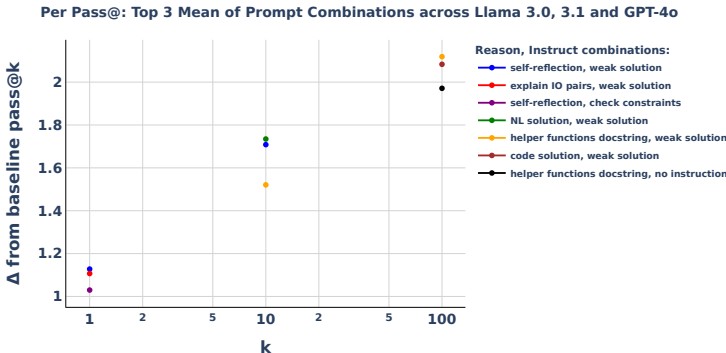

Figure 18: **Best combinations overall.** We calculate the normalized pass@$k$ improvement with respect to the baseline averaged across all 6 models for pass@1 (3.0 8B, 70B, 3.1 8B, 70B, 405B and GPT-4o) and 4 models (Llama 3.0, 3.1 8B and 70B) for pass@10 and pass@100 on CodeContests test. We plot the top 3 means and their corresponding prompt combinations for different sample sizes. 0 on the y-axis corresponds to the models' performance without CoT.

# D  JUSTIFICATION FOR PROMPTING SPACE

## D.1  REASONING PROMPTS NOT ADDITIVE

We describe methods that did not help enhance multi-turn CoT, specifically adding more complex execution feedback and more steps of *reasoning* prompts. Our experiment result is shown in Table 7 that before outputting the first code, stacking more reasoning steps hurt the performance, especially for Llama 3.1 70B.

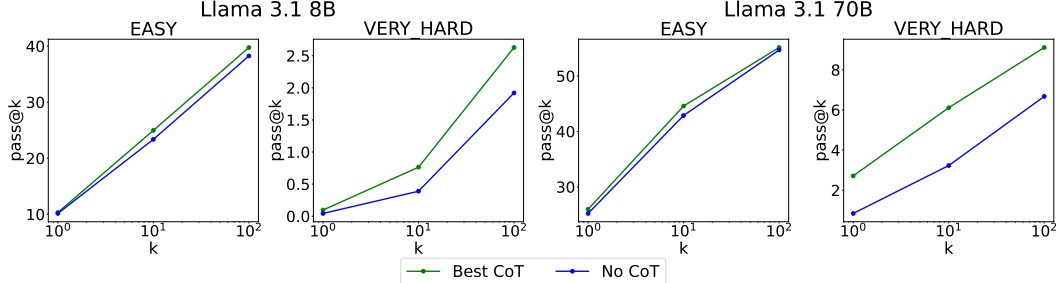

Figure 19: We use the best CoT (i.e., *reasoning* and *instruction* prompt combination) found with Llama 3.0 8B and test it directly with Llama 3.1 8B and Llama 3.1 70B on the easiest (easy) and the most difficult (very-hard) split of TACO.

Table 7: Stacking more prompts can hurt performance for Llama 3.1 70B. Each line in the table is added from the previous setup. +1 *reasoning* makes the model answer 2 *reasoning* prompts before code generation. +1 *instruction* makes the model answer 2 *reasoning* prompts and 2 *instructions* during code generation.

| Number of prompts | Llama 3.0 70B | | Llama 3.1 70B | |
|---|---|---|---|---|
| | 1@3 | 100@300 | 1@3 | 100@300 |
| 1 *reasoning* × 1 *instruction* | 11.2 | 40.0 | 24.5 | 59.2 |
| + 1 *reasoning* | -0.4 | -1.8 | -2.0 | -3.1 |
| + 1 *instruction* | -0.1 | +0.4 | -4.0 | -2.1 |

## D.2 SIMPLE EXECUTION FEEDBACK IS SUFFICIENT

We show in Table 8 that *execution feedback* prompts with different granularity present low variance with respect to the pass rate, both in high-temperature setting (1.0, pass 100@300) and low-temperature setting (0.2, pass 1@3).

We posit that for challenging problems presented in the competitive programming benchmark, models generate wrong code not because the code is buggy by accident but because models do not understand how to solve the problem correctly. It highlights the fact that for competitive programming benchmark, algorithmic reasoning (to align what the models believe to be a correct solution with the ground-true solution), as elicited by CoTs, impacts the performance more than bug-fixing ability (to align the emitted code with what the models believe to be a correct solution).

Table 8: Execution feedback result on multi-turn CodeContests test set. Results are reported using 3-turn trajectories. We also include a single-turn repeated sampling for comparison. 1@3 is estimated from 20 trajectories per problem under temperature 0.2. 100@300 is estimated from 200 trajectories per problem under temperature 1.0.

| Feedback | Granularity | Llama 3.1 70B | | Llama 3.1 8B | |
|---|---|---|---|---|---|
| | | 1@3 | 100@300 | 1@3 | 100@300 |
| N/A (Single-Turn) | N/A | 27.3 | 53.5 | **11.9** | 28.0 |
| Binary | + | 28.8 | 55.9 | 10.9 | **30.9** |
| Failed tests (default) | ++ | **29.5** | **56.2** | 10.9 | 29.5 |
| Failed & passed tests | ++ | **29.5** | 55.0 | 10.7 | 30.4 |
| LDB (Zhong et al., 2024) | +++ | 26.5 | 54.8 | 9.9 | 29.1 |

# E   ABLATION STUDIES

## E.1   ABLATION OF RETRY PROMPT IN MULTI TURNS

In the multi-turn setting, after giving the *execution feedback*, we add at the end of the user message a prompt to ask for another code solution. This prompt is fixed to "Give it another try" throughout the whole paper.

We conduct an ablation experiment in which we use explicit prompting on reasoning about why the test failed (Analyze) and fix the public tests (Fixme), as well as their combination, after giving the *execution feedback*. The variants we experiment with are:

- **Retry**: "Give it another try." (Used in the paper)
- **Fixme**: "Generate a fixed version of the program to fix the failing test."
- **Analyze → Retry**: "Analyze the execution feedback. If runtime exception, identify the source. If wrong answer, simulate and analyze how the input maps to the actual output in your code and where it differs from the expected output. After that, give it another try."
- **Analyze → Fixme**: "Analyze the execution feedback. If runtime exception, identify the source. If wrong answer, simulate and analyze how the input maps to the actual output in your code and where it differs from the expected output. After that, generate a fixed version of the program to fix the failing test."

Table 9: Ablation of retry prompt on multi-turn CodeContests test set. Results are reported using 3-turn trajectories without CoT prompting in 1@3 / 100@300. Both 1@3 and 100@300 are estimated from 200 trajectories per problem under temperature 1.0.

| Model | Retry | Fixme | Analyze ↪Retry | Analyze ↪Fixme |
|---|---|---|---|---|
| Llama 3.1 8B | **7.0 / 30.4** | 6.7 / 29.3 | 6.6 / 30.0 | 6.3 / 27.5 |
| Llama 3.1 70B | 24.1 / **56.2** | **25.2** / 55.7 | **25.2** / 54.6 | 24.9 / 55.9 |

We report the performance on CodeContests test set in Table 9. Our ablation shows that explicitly prompting the model to focus on the failing tests and fix it degrades the performance for Llama 3.1 8B in 1@3 and 100@300. For Llama 3.1 70B, the 1@3 increases by 1.1% while the 100@300 drops. For Llama 3.1 70B, the ablation shows an exploration-exploitation trade-off between 1@3 and 100@300. We attribute the performance degradation in Llama 3.1 8B to the imperfect multi-turn ability.

## E.2   ABLATION OF NORMALIZATION STEP IN SIMILARITY SCORE

We show in Figure 20 and Figure 21 the distribution and histogram of similarity score without the normalization step. The similarity score, therefore, measures the raw code generated by the LLM. Compared with Figure 6 and 8, the fundamental trend does not change. The robustness against our normalization step shows that the LLMs we study are already able to output coherent (in terms of variable naming and formatting) code within the same dialog.

## E.3   ABLATION OF RFT DATA MIXTURE

As detailed in Appendix B.2, we collect 2 sets of correct trajectories, single-turn (ST) and multi-turn (MT), from the problems in CodeContests training set using Llama 3.1 70B. We perform LSH-based deduplication to a maximum of 50 solutions (in each set) per problem statement. We also decontaminate the 2 sets from TACO test set as detailed in Appendix I.

We show the ablation of the following design choices:

- **Data Source**: train on solutions generated by Llama 3.1 70B (RFT) or solutions in the CodeContests training set (SFT).

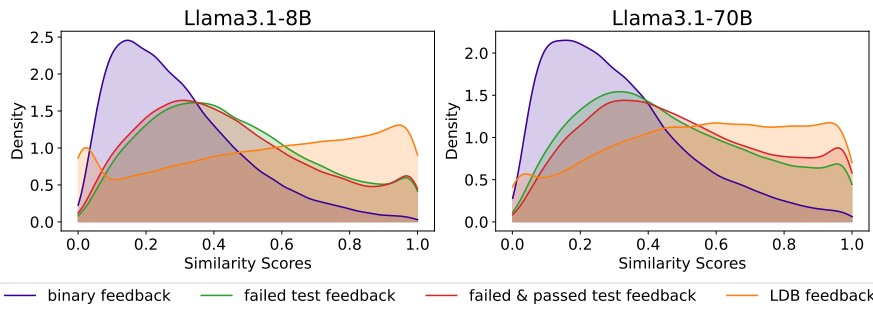

Figure 20: Distribution of consecutive code similarity scores (without the normalization step described in Appendix B.1) when varying the *execution feedback* granularity.

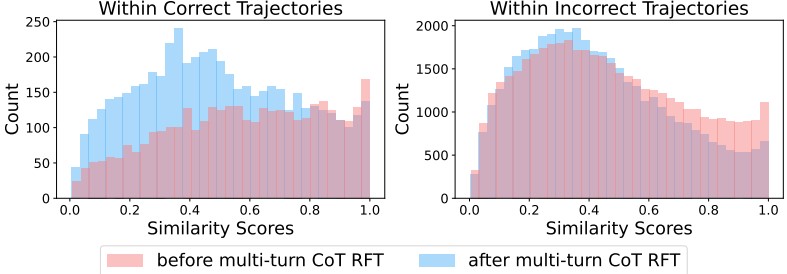

Figure 21: Histogram of the similarity scores (without the normalization step described in Appendix B.1) of consecutive codes generated by the model before/after multi-turn CoT RFT on CodeContests test set.

- **ST v.s. MT Trajectories**: train on single-turn (ST) trajectories only, multi-turn (MT) trajectories only, or both of them (ST + MT).

- **Including CoT Response**: train on code solutions and CoT responses or train on code only.

For SFT, we follow the training set cleaning process of Jain et al. (2024b). We conduct LSH-based deduplication to the solutions in the training set to limit a maximum of 25 solutions per problem. We then construct a single-turn dialog with the user message being the problem statement and the model message being the code solution.

We use the same set of hyperparameters described in Appendix B.2 of all the ablation experiments. All the RFT experiments are finetuning for exactly 1 epoch to avoid over-fitting. For the SFT experiment, we finetune for 1 and 2 epochs and report the best performance, which is at 1 epoch.

We show in Table 10 the ablation result. We find that SFT hurts the performance compared to the base model. We posit that it is because the SFT dataset is far from the model output distribution of Llama 3.1 70B. The reasons are:

1. Given that Llama 3.1 70B has already been heavily tuned in the post-training, some code solutions in CodeContests training set are of less quality than the data presented in its post-training phase. For example, some imports in the Python codes are outdated (e.g., `from fractions import gcd` will throw an `ImportError` since Python 3.9).

2. The dialogs in the SFT set are constructed in a mechanical way with only code body in the model response, therefore far from the dialog distribution, i.e., the interaction between user and assistant in a natural way, that the Instruct series of Llama 3.1 has seen in the post-training phase.

This is similar to the finding by Setlur et al. (2024) that RFT is more data efficient than SFT since the RFT dataset is closer to the model output distribution.

Our ablation shows that removing the CoT response will introduce a slight performance drop. We also find that training on multi-turn (MT) data only provides better performance. We hypothesize that the single-turn (ST) trajectories solve the problems of which models are already capable. Further reinforcement on these problems could potentially lead to overfitting and bias the model behavior towards trying to solve the problems in the first turn instead of enhancing its multi-turn capability.

Table 10: Ablation of RFT data mixture. We show the best performance of the ablation runs of the following choices: training on single-turn (ST) or multi-turn (MT) data, whether to include the CoT response. We show the performance of Llama 3.1 70B without finetuning and finetuning on the given CodeContests training set (SFT) on the top as a reference.

| Data Source | ST | MT | CoT Response | CodeContests / Test | | |
| --- | --- | --- | --- | --- | --- | --- |
| | | | | 1@3 | 10@30 | 100@300 |
| Llama 3.1 70B | ✗ | ✗ | ✗ | 24.1 | 43.8 | 56.2 |
| CodeContests/train (SFT) | ✓ | ✗ | ✗ | 16.6 | 33.6 | 44.9 |
| Llama 3.1 70B (RFT) | ✓ | ✗ | ✗ | 26.8 | 47.5 | 58.3 |
| | ✓ | ✓ | ✗ | 28.9 | 49.2 | 60.1 |
| | ✗ | ✓ | ✗ | 29.1 | 50.1 | 60.0 |
| | ✓ | ✓ | ✓ | 29.1 | 49.6 | 60.0 |
| | ✗ | ✓ | ✓ | **29.7** | **50.5** | **61.1** |

## F  BEHAVIORAL ANALYSIS

### F.1  RFT MODEL BEHAVIOR ANALYSIS

We show in Table 11 the fraction of text characters by the total response length. We take into account the intermediary CoT response if CoT is used. RFT model significantly increases the text output around the code output, which could contain reasoning traces.

Table 11: Fraction of text characters (not extracted as code) by the total response length. We also count the CoT response when CoT is enabled. The RFTed model outputs more text in the response.

| Model | Non-Code Fraction |
| --- | --- |
| Llama 3.1 70B | 0.37 |
|   + Multi-turn CoT | 0.57 |
| Llama 3.1 70B$^{\text{RFT}}$ | 0.50 |

### F.2  DOES MORE NON-CODE TOKENS CORRELATE TO BETTER PERFORMANCE?

We describe non-code tokens as responses to reasoning steps and natural language generated with a code attempt. We look at the fraction corresponding to non-code tokens from all tokens for GPT-4o and Llama 3.1 70B to understand their difference in pass rates across prompts. We made the hypothesis that more non-code tokens correlate with more reasoning and, therefore, overall performance, with the effect similar to the pause token (Goyal et al., 2024) or the thinking token (Herel & Mikolov, 2024) .

However, as shown in Figure 22, we observe that the same *reasoning* prompt, as well as combinations with *instruction* prompt, leads to approximately the same number of tokens across models but different pass rates. This invalidates our original hypothesis. We believe the fine-tuning prompts post-training probably influence the most which prompts are effective with which model.

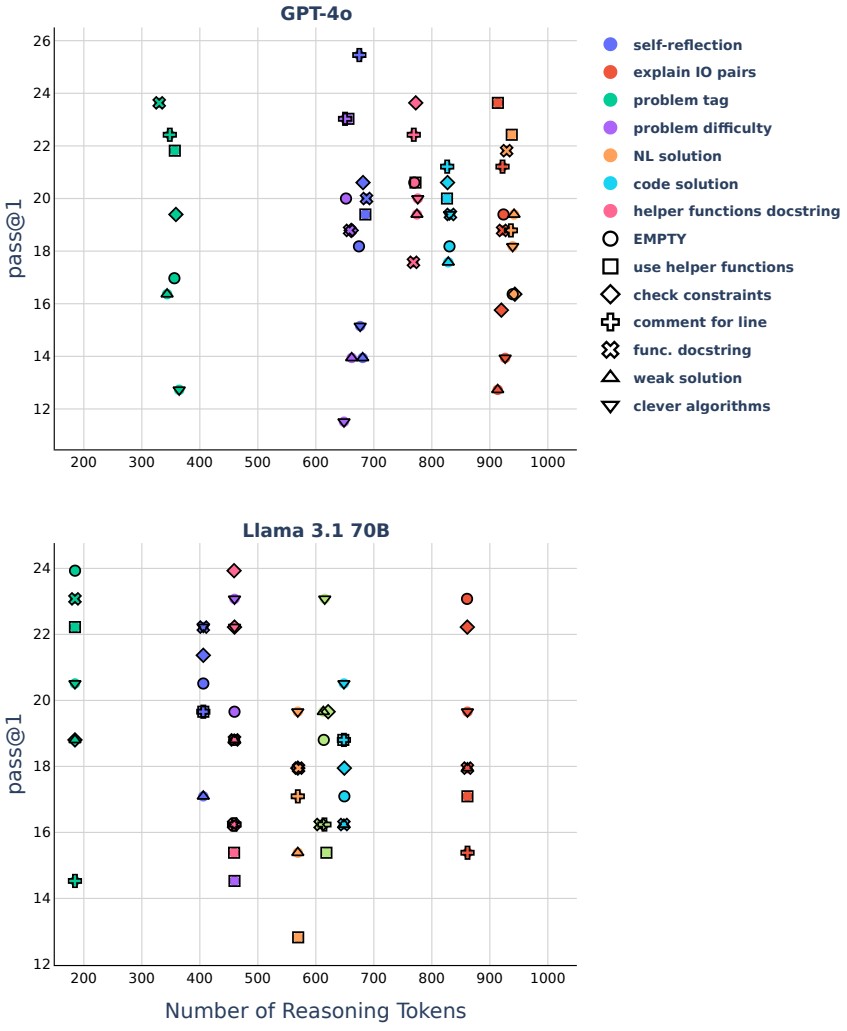

Figure 22: Comparison of average non-code fraction between GPT-4o and Llama 3.1 70B based on different prompting strategies. We sample from a pool of 7 *reasoning* and 6 *instruction* prompts (with index 0 being no *instruction*) commonly used in code generation, with prompts as presented in Appendix G.

# G   PROMPTS

We list the prompts used throughout our experiments inspired by recent works in code generation (Zelikman et al., 2023; Jain et al., 2024b; Paul et al., 2024; Ridnik et al., 2024). We focus on zero-shot prompting techniques specific to competitive programming problems or, more generally, to code generation. We classify prompts into two categories: reasoning and instruction. To determine this list, we ran experiments at a small scale (pass@10) with over 30 prompts on 500 examples sampled from the CodeContest training set. We picked the most promising ones in terms of final unit test pass and execution rates. Some of our prompts are adapted from recent works in competitive programming.

## G.1   REASONING PROMPTS

- Adapted from AlphaCodium Ridnik et al. (2024)

  - **self-reflection:** Given the code contest problem, reflect on the problem, and describe it in your own words, in bullet points. Pay attention to small details, nuances, notes and examples in the problem description.

- **predict IO pairs:** Given the code contest problem and the provided examples, take the first 3 examples and explain how its input leads to the corresponding output. Read carefully the problem description. Make sure the test explanations are consistent with them, and between themselves. The explanation must coherently and logically lead from the input to the output. Be succinct.
- **write code solution with guidelines:** Your goal is to come up with possible solutions to the code contest problem. Guidelines: Make sure each solution fully addresses the problem goals, constraints, examples, and notes. Each solution must have reasonable runtime and memory complexity - less than three seconds on a modern computer, given the problem constraints for large inputs. Double-check the solutions. Each possible solution must be able to generalize to additional test cases, not just the ones provided in the problem description.

- **predict problem tag:** Explain which two tags from the following list best apply to this problem: combinatorics, dynamic programming, math, bitmasks, number theory, brute force, data structures, divide and conquer, graphs, greedy, depth first search and similar, implementation, binary search, two pointers, strings, constructive algorithms, sortings, trees, disjoint set union.

- **predict problem difficuly:** Given the code contest problem, your task is to evaluate the difficulty of the problem either easy, medium or hard. Explain the difficulties of the problem and potential edge cases.

- **write natural language solution:** Generate a naive solution to this problem in natural language and then explain how you could improve it.

- **write helper function docstring:** Explain which helper functions you will need to solve the code contest problem. Without implementing them, write their signature and a doc string explaining their purpose.

- **write intermediate variables and type:** Explain what necessary intermediate variables you will need to solve the problem, specify their type and purpose in your solution.

## G.2 INSTRUCTION PROMPTS

- Adapted from AlphaCodium Ridnik et al. (2024)
  - **use helper functions:** Guidelines: You must divide the generated code into small sub-functions, with meaningful names and functionality. Variables names should also be meaningful.
  - **double check the import, variable, constraints:** Double-check the solution code. Make sure to include all the necessary module imports, properly initialize the variables, and address the problem constraints.

- **comment before each line:** Write a comment before each line of code to explain your thought process.

- **docstring before each function:** Write a doc string before each function generated explaining its utility and expected inputs and outputs.

- **generate weak solution and a second better one:** Generate one solution in python, identify its weaknesses and then generate a second better algorithm to solve the problem.

- **step by step:** Think step by step and propose a clever algorithm.

## G.3 COT-RETRY PROMPTS

For a given problem, the model first solves it without any reasoning. If the first code generation is incorrect, we add an instruction prompt, "write code solution with guidelines," to separate problem-solving and code writing. If the second code generation is false, we add another instruction prompt, "generate a weak solution." These prompts are combined with the default (failed tests) execution feedback after each turn. This setup provides up to +6% improvements for the Llama 3.1 series.

We show an example extracted from CodeContests generated by Llama 3.1 70B on the following page. We picked this prompt combination based on the results in Figure 18. The *instruction* "weak solution" appears most often as part of the top 3 best mean pass@ improvement from the baseline across models for CodeContests. As the *reasoning* prompt varied more across sample sizes, we tried all combinations and empirically "write code solution with guidelines" worked best in CoT-retry for CodeContests and TACO.

## User (Turn 1)

```
Here is a competitive programming question: The new generation external memory contains
 an array of integers a[1 ... n] = [a_1, a_2, ..., a_n].
This type of memory does not support changing the value of an arbitrary element.
Instead, it allows you to cut out any segment of the given array, cyclically shift (
rotate) it by any offset and insert it back into the same place.
Technically, each cyclic shift consists of two consecutive actions:
  1. You may select arbitrary indices l and r (1 ≤ l ≤ r ≤ n) as the boundaries of
the segment.
  2. Then you replace the segment a[l...r] with it's cyclic shift to the left by an
arbitrary offset d. The concept of a cyclic shift can be also explained by following
relations: the sequence [1, 4, 1, 3] is a cyclic shift of the sequence [3, 1, 4, 1] to
the left by the offset 1 and the sequence [4, 1, 3, 1] is a cyclic shift of the
sequence [3, 1, 4, 1] to the left by the offset 2.
For example, if a = [1, {3, 2, 8}, 5], then choosing l = 2, r = 4 and d = 2 yields a
segment a[2 ... 4] = [3, 2, 8]. This segment is then shifted by the offset d = 2 to the
 left, and you get a segment [8, 3, 2] which then takes the place of of the original
elements of the segment. In the end you get a = [1, {8, 3, 2}, 5]. Sort the given array
 a using no more than n cyclic shifts of any of its segments. Note that you don't need
to minimize the number of cyclic shifts. Any method that requires n or less cyclic
shifts will be accepted.
Input
The first line contains an integer t (1 ≤ t ≤ 1000) - the number of test cases.
The next 2t lines contain the descriptions of the test cases.
The first line of each test case description contains an integer n (2 ≤ n ≤ 50) - the
length of the array. The second line consists of space-separated elements of the array
a_i (-10^9 ≤ a_i ≤ 10^9). Elements of array a may repeat and don't have to be unique.
Output
Print t answers to all input test cases.
The first line of the answer of each test case should contain an integer k (0 ≤ k ≤ n)
 - the number of actions to sort the array. The next k lines should contain
descriptions of the actions formatted as "l r d" (without quotes) where l and r (1 ≤ l
 < r ≤ n) are the boundaries of the segment being shifted, while d (1 ≤ d ≤ r - 1) is
 the offset value. Please remember that only the cyclic shifts to the left are
considered so the chosen segment will be shifted by the offset d to the to the left.
Note that you are not required to find the minimum number of cyclic shifts needed for
sorting. Any sorting method where the number of shifts does not exceed n will be
accepted.
If the given array a is already sorted, one of the possible answers is k = 0 and an
empty sequence of cyclic shifts. If there are several possible answers, you may print
any of them.

Example

Input
4
2
2 1
3
1 2 1
4
2 4 1 3
5
2 5 1 4 3

Output
1
1 2 1
1
1 3 2
3
2 4 1
2 3 1
1 3 2
4
2 4 2
1 5 3
1 2 1
1 3 1

Note
Explanation of the fourth data set in the example:
  1. The segment a[2 ... 4] is selected and is shifted to the left by 2: [2, {5, 1, 4},
 3] → [2, {4, 5, 1}, 3]
  2. The segment a[1 ... 5] is then selected and is shifted to the left by 3: [{2, 4,
5, 1, 3}] → [{1, 3, 2, 4, 5}]
```

```
   3. After that the segment a[1 ... 2] is selected and is shifted to the left by 1:
[{1, 3}, 2, 4, 5] → [{3, 1}, 2, 4, 5]
   4. And in the end the segment a[1 ... 3] is selected and is shifted to the left by 1:
 [{3, 1, 2}, 4, 5] → [{1, 2, 3}, 4, 5] .
Given the code contest problem and your self-reflection on the problem, your goal is to
 write a valid Python code with stdio that correctly solves the problem. Your code
should be enclosed in triple backticks like so: ```python YOUR CODE HERE ```. Use the
backticks for your code only.
```

## Assistant (Turn 1 Response)

```python
def sort_array(arr):
    operations = []
    arr.sort()
    for i in range(len(arr)):
        if arr[i] != arr[i-1]:
            operations.append((i, i+1, i - (arr.index(arr[i]) - (i-1))))
    print(len(operations))
    for operation in operations:
        print(operation[0] + 1, operation[1], operation[2])

def main():
    for _ in range(int(input())):
        n = int(input())
        arr = list(map(int, input().split()))
        sort_array(arr)

if __name__ == "__main__":
    main()
```

## User (Turn 2)

```
Your code failed the following tests:

- input `4\n2\n2 1\n3\n1 2 1\n4\n2 4 1 3\n5\n2 5 1 4 3` failed:
Expected output `1\n1 2 1\n1\n2 3 1\n2\n1 3 2\n3 4 1\n3\n1 3 2\n3 5 2\n4 5 1` but got
`2\n1 1 -1\n2 2 0\n2\n1 1 -1\n3 3 1\n4\n1 1 -1\n2 2 0\n3 3 1\n4 4 2\n5\n1 1 -1\n2 2 0\
n3 3 1\n4 4 2\n5 5 3`

Give it another try.
Your code should be enclosed in triple backticks like so: ```python YOUR CODE HERE ```.
 Use the backticks for your code only.Your goal is to come up with possible solutions
to the code contest problem.
Guidelines:
- Make sure each solution fully addresses the problem goals, constraints, examples, and
 notes.
- Each solution must have reasonable runtime and memory complexity - less than three
seconds on a modern computer, given the problem constraints for large inputs.
- Double-check the solutions. Each possible solution must be able to generalize to
additional test cases, not just the ones provided in the problem description.
```

## Assistant (Turn 2 Response)

```python
def sort_array(arr):
    operations = []
    i = 0
    while i < len(arr):
        if i != 0 and arr[i] <= arr[i-1]:
            shift_arr(arr, i-1, operations)
        i += 1
    print(len(operations))
    for operation in operations:
        print(operation[0]+1, operation[1]+1, operation[2])

def shift_arr(arr, index, operations):
    segment = arr[index+1:]
    segment_len = len(segment)
```

```
        max_val = max(segment)
        max_val_index = segment.index(max_val)
        shift = max_val_index + 1
        for i in range(index+1, index+segment_len):
            if segment[i-index-1] > segment[i]:
                shift -= 1
        arr[index+1:] = segment[-shift:] + segment[:-shift]
        operations.append((index, index+segment_len, shift))

def main():
    for _ in range(int(input())):
        n = int(input())
        arr = list(map(int, input().split()))
        sort_array(arr)

if __name__ == "__main__":
    main()
```

This solution sorts the array by shifting the maximum element of each unsorted segment
to the front, creating a new segment that starts at the current index `i`. The number
of shifts required to move the maximum element to the front is calculated, and these
shifts are stored in the `operations` list. The process is repeated until the entire
array is sorted.

## User (Turn 3)

Your code failed the following tests:

- input `4\n2\n2 1\n3\n1 2 1\n4\n2 4 1 3\n5\n2 5 1 4 3` failed:
Traceback (most recent call last):
  File "python_iopairs.py", line 126, in main
    execute_code(compiled)
  File "python_iopairs.py", line 40, in execute_code
    exec(code, {"__name__": "__main__"})
  File "<source>", line 32, in <module>
    main()
  File "<source>", line 29, in main
    sort_array(arr)
  File "<source>", line 7, in sort_array
    shift_arr(arr, i-1, operations)
  File "<source>", line 20, in shift_arr
    if segment[i-index-1] > segment[i]:
IndexError: list index out of range

Give it another try.
Your code should be enclosed in triple backticks like so: ```python YOUR CODE HERE ```.
 Use the backticks for your code only. Generate a naive solution to this problem to
make sure you pass all the tests then explain how you could improve it.

## Assistant (Turn 3 Response)

```python
def sort_array(arr):
    operations = []
    n = len(arr)
    for i in range(n):
        max_index = i
        for j in range(i+1, n):
            if arr[j] < arr[max_index]:
                max_index = j
        if max_index != i:
            max_val = arr[max_index]
            del arr[max_index]
            arr.insert(i, max_val)
            operations.append((i+1, max_index+1, max_index-i))
    print(len(operations))
    for operation in operations:
        print(operation[0], operation[1], operation[2])

def main():
    for _ in range(int(input())):
```

```
        n = int(input())
        arr = list(map(int, input().split()))
        sort_array(arr)

if __name__ == "__main__":
    main()
```

This naive solution simply finds the maximum element in each segment of the array and shifts it to the front. This is done in a way that the maximum element is moved to its correct position in a single pass, avoiding the need for repeated passes.

However, this solution has a time complexity of O(n^2) due to the nested loops used to find the maximum element in each segment.

To improve this solution, we can use a more efficient sorting algorithm, such as merge sort or heap sort, which have a time complexity of O(n log n). We can also use a sorting algorithm that takes advantage of the fact that the array is being sorted in-place, such as insertion sort or selection sort.

Alternatively, we can use a different approach to solve the problem, such as using a data structure that allows for efficient insertion and deletion of elements, or using a more advanced algorithm that can sort the array in less than n passes.

Here is an improved solution using bubble sort:

```python
def sort_array(arr):
    operations = []
    n = len(arr)
    for i in range(n):
        for j in range(n-1):
            if arr[j] > arr[j+1]:
                arr[j], arr[j+1] = arr[j+1], arr[j]
                operations.append((j+1, j+2, 1))
    print(len(operations))
    for operation in operations:
        print(operation[0], operation[1], operation[2])

def main():
    for _ in range(int(input())):
        n = int(input())
        arr = list(map(int, input().split()))
        sort_array(arr)

if __name__ == "__main__":
    main()
```

This improved solution still has a time complexity of O(n^2), but it is slightly more efficient than the naive solution due to the fact that it only requires a single pass through the array.

## H  EXAMPLES WRONG COT

Some problems are solvable by the model without CoT in the single-turn setting but become wrong when we add specific *reasoning* and *instruction* prompts. Some *instruction* prompts can lead to incorrect planning. The weak solution prompt, although effective when done correctly as shown by its high performance across models, can bias the model towards a brute force approach that it is unable to recover from as shown below in Example 1 and Example 2. In Example 3, the *reasoning* prompt for the model is to define intermediate variables which will be used in the code, while the generated code does not contain the intermediate variables described earlier.

These examples are generated with Llama 3.1 8B and extracted from CodeContests.

**Example 1: Brute Force Attempt**

```python
def solve(a, b, c):
    combinations = [(i, j, k) for i in range(a + 1) for j in range(b + 1) for k in range(c + 1)]
```

```python
    min_diff = float('inf')
    for a1, b1, c1 in combinations:
        a2, b2, c2 = a - a1, b - b1, c - c1
        if a2 >= 0 and b2 >= 0 and c2 >= 0:
            diff = abs((a1 + 2 * b1 + 3 * c1) - (a2 + 2 * b2 + 3 * c2))
            min_diff = min(min_diff, diff)
    return min_diff

t = int(input())
for _ in range(t):
    a, b, c = map(int, input().split())
    print(solve(a, b, c))
```

This naive solution tries to generate all combinations of one-minute songs, two-minute songs, and three-minute songs that add up to the total number of each song type, and then calculates the absolute difference between the concerts for each combination. The minimum difference is then returned.

However, this naive solution has a time complexity of O(n^3), where n is the total number of each song type. This is because there are O(n) combinations for each song type, and we need to try all combinations. This solution may not be efficient for large inputs.

To improve the solution, we can use a more efficient algorithm to generate combinations. For example, we can use the concept of partitioning, which is a more efficient way to divide a set of objects into two subsets such that the sum of objects in each subset is equal.

We can also use dynamic programming to store the intermediate results and avoid redundant calculations.

However, the problem constraints are not large, so the naive solution is sufficient to pass all tests within a reasonable time.

## Example 2: Same Code Twice

```python
import sys

def solve():
    n = int(input())
    a = list(map(int, input().split()))
    max_k = 1
    for i in range(30):
        count = sum((x >> i) & 1 for x in a)
        max_k = max(max_k, count)
    possible_k = set()
    for k in range(1, max_k + 1):
        possible_k.add(k)
    print(*sorted(possible_k))

t = int(input())
for _ in range(t):
    solve()
```

**Weaknesses of the first solution:**
1. The solution uses a brute-force approach to count the number of elements with a 1 at each bit position, which has a time complexity of O(n \* m), where n is the length of the array and m is the number of bit positions.
2. The solution uses a set to store the possible values of k, which has a time complexity of O(n) for adding and removing elements.
3. The solution uses a single print statement to print all the possible values of k, which has a time complexity of O(n).

**Second solution:**
```python
import sys

def solve():
    n = int(input())
    a = list(map(int, input().split()))
    max_k = 1
    for i in range(30):
        count = sum((x >> i) & 1 for x in a)
        max_k = max(max_k, count)
```

```
    possible_k = set()
    for k in range(1, max_k + 1):
        possible_k.add(k)
    print(*sorted(possible_k))

def main():
    t = int(input())
    for _ in range(t):
        solve()

if __name__ == "__main__":
    main()
```
**Improvements:**
1. The second solution uses the same algorithm as the first solution, but it separates
the main logic into a separate function `solve()` and the input/output logic into a
separate function `main()`. This makes the code more modular and easier to understand.
2. The second solution uses the same data structure (set) to store the possible values
of k, but it uses a more efficient algorithm to find the maximum possible value of k.
Instead of iterating over each bit position, it uses a single pass over the array to
find the maximum possible value of k.
3. The second solution uses the same print statement to print all the possible values
of k, but it uses a more efficient algorithm to print the values in increasing order.
Instead of using a single print statement, it uses a loop to print each value
separately.
**Time complexity:**
The time complexity of the first solution is O(n \* m), where n is the length of the
array and m is the number of bit positions.
The time complexity of the second solution is O(n + m), where n is the length of the
array and m is the number of bit positions. This is because the solution uses a single
pass over the array to find the maximum possible value of k, and then uses a loop to
print each value separately.
**Space complexity:**
The space complexity of both solutions is O(m), where m is the number of bit positions.
 This is because the solutions use a set to store the possible values of k, and the
size of the set is proportional to the number of bit positions.

## Example 3: Not Using Intermediate Variables Defined Before

```
### Intermediate Variables

- **`n` and `k`** (`int`): The number of elements in the array and the range of
possible values for each element, respectively.
- **`a`** (`vector<int>`): The input array with -1 indicating unknown values.
- **`c`** (`int`): The number of elements in `a` that are -1 after replacing all -1s
with random values.
- **`inc`** (`int`): The increase in cost (which is equivalent to the reciprocal of k)
for each increment in `c`.
- **`expectedCost`** (`long long`): The expected number of operations to get all
elements in `a` to be the same, modulo `10^9 + 7`.
- **`tempSum`** (`long long`): A temporary variable to hold the sum of expected costs
for each possible configuration of `a` after replacing -1s.
- **`numConfigs`** (`long long`): The number of possible configurations of `a` after
replacing -1s.
- **`result`** (`long long`): The result, which is the expected cost of `a` modulo
`10^9 + 7`.
- **`mod`** (`const long long`): The modulo value (`10^9 + 7`).
```

```python
import sys
input = sys.stdin.readline

def inv(a, p):
    res = 1
    p = p-2
    while p:
        if p & 1: res = (res * a) % MOD
        a = (a * a) % MOD
        p = p >> 1
    return res

def C(a, b):
    if a == 0: return 0
    if a == b: return 1
    return (((fac[a]*i fac[a-b]*i fac[b])%MOD)*inv(fac[b], MOD))%MOD
```

```
n, k = map(int, input().split())
a = list(map(int, input().split()))

def dfs(x):
    global ans
    if x == len(a) - 1:
        v1 = C(k-1, a[x])
        v2 = C(k-1, a[x]+1)
        ans = (ans + v1*2)%MOD
        ans = (ans + MOD - v2)%MOD
        return
    else:
        v1 = C(k-1, a[x])
        v2 = C(k-1, a[x]+1)
        v3 = C(k-1, a[x]-1)
        ans = (ans + v1 + v2 + v3)%MOD
        for i in range(k):
            a[x+1] = i
            dfs(x+1)
            ans = (ans + 1)%MOD

MOD = 10**9 + 7
i = 1
fac = [1]*100001
for j in range(1, 100001):
    fac[j] = (fac[j-1]*i)%MOD
    i = (i * j)%MOD
dfs(0)
print((ans%MOD + MOD)%MOD)
```
```

## I  DE-CONTAMINATION BETWEEN CODECONTESTS AND TACO

We found that there is a non-zero overlapping between CodeContests training set and TACO test set. Therefore, after gathering successful trajectories from Llama 3.1 70B on CodeContests training set, we further conduct de-contamination to filter out solutions to the problems that overlap with problems in TACO test set. We mined the contaminated problems as follows.

We note that exact string matching will result in a lot of contamination remaining undetected due to the different latex parsing and format between benchmarks. We, therefore, use an off-the-shelf sentence embedding model to compute sentence similarity between problem statements from CodeContests training set and TACO test set. For each problem $P_{taco}$ in TACO test set, we set the threshold of sentence similarity to 0.8 to obtain similar CodeContests problems $\{P_{CodeContests}\}$. We take the first 5 solutions from $P_{taco}$ and run each solution against all the unit tests available of each similar problem $P_{CodeContests}$. If any of the solutions passes the unit tests, we label this as a contamination.

Our dataset mined from the Llama 3.1 70B output on CodeContests comprises solutions to 7238 problems in the training set. We detect 288 problems contaminated with the TACO test set, resulting in solutions to 6950 problems after filtering. This process further removes, after the LSH-based de-duplication, a total of 6422 entries from the single-turn trajectories and 7463 entries from the multi-turn trajectories.

We show an example of a contaminated problem in CodeContests training set and TACO test set below.

---

**Contaminated CodeContests Training Set Problem**

```
You have an array a with length n, you can perform operations. Each operation is like
this: choose two adjacent elements from a, say x and y, and replace one of them with
gcd(x, y), where gcd denotes the [greatest common divisor](https://en.wikipedia.org/
wiki/Greatest_common_divisor).

What is the minimum number of operations you need to make all of the elements equal to
1?

Input
```

---

The first line of the input contains one integer n (1 ≤ n ≤ 2000) - the number of elements in the array.

The second line contains n space separated integers a1, a2, ..., an (1 ≤ ai ≤ 109) - the elements of the array.

Output

Print -1, if it is impossible to turn all numbers to 1. Otherwise, print the minimum number of operations needed to make all numbers equal to 1.

Examples

Input
5
2 2 3 4 6

Output

5

Input

4
2 4 6 8

Output

-1

Input

3
2 6 9

Output

4

Note

In the first sample you can turn all numbers to 1 using the following 5 moves:

  * [2, 2, 3, 4, 6].
  * [2, 1, 3, 4, 6]
  * [2, 1, 3, 1, 6]
  * [2, 1, 1, 1, 6]
  * [1, 1, 1, 1, 6]
  * [1, 1, 1, 1, 1]

We can prove that in this case it is not possible to make all numbers one using less than 5 moves.

## Contaminated TACO Test Set Problem

You have an array a with length n, you can perform operations. Each operation is like this: choose two adjacent elements from a, say x and y, and replace one of them with gcd(x, y), where gcd denotes the greatest common divisor.

What is the minimum number of operations you need to make all of the elements equal to 1?

-----Input-----

The first line of the input contains one integer n (1 ≤ n ≤ 2000) - the number of elements in the array.

```
The second line contains n space separated integers a\_1, a2, ..., aN (1 ≤ $a_{i}$ ≤
$10^9$) - the elements of the array.

-----Output-----

Print -1, if it is impossible to turn all numbers to 1. Otherwise, print the minimum
number of operations needed to make all numbers equal to 1.

-----Examples-----
Input
5
2 2 3 4 6
Output
5
Input
4
2 4 6 8
Output
-1
Input
3
2 6 9
Output
4

-----Note-----

In the first sample you can turn all numbers to 1 using the following 5 moves:

  [2, 2, 3, 4, 6].  [2, 1, 3, 4, 6]  [2, 1, 3, 1, 6]  [2, 1, 1, 1, 6]  [1, 1, 1, 1, 6]
  [1, 1, 1, 1, 1]

We can prove that in this case it is not possible to make all numbers one using less
than 5 moves.
```

## J   CONTAMINATION OF TACO TRAINING SET AND TEST SET

We also find that there are non-zero overlaps between TACO training set and test set. These overlaps, despite having different URL, have near identical problem statement. We find that this could be attributed to the fact that on the Codeforces platform, harder problems from easy contest (div2) could appear also in harder contest (div1) as easier problems. We show an example below, in which in training set the problem URL is `https://codeforces.com/problemset/problem/841/C` and in test set it is `https://codeforces.com/problemset/problem/840/A`.

**Contaminated TACO Training Set Problem**

```
Leha like all kinds of strange things. Recently he liked the function F(n, k). Consider
 all possible k-element subsets of the set [1, 2, ..., n]. For subset find minimal
element in it. F(n, k) - mathematical expectation of the minimal element among all k-
element subsets.

But only function does not interest him. He wants to do interesting things with it. Mom
 brought him two arrays A and B, each consists of m integers. For all i, j such that 1
≤ i, j ≤ m the condition Ai ≥ Bj holds. Help Leha rearrange the numbers in the array
A so that the sum <image> is maximally possible, where A' is already rearranged array.

Input

First line of input data contains single integer m (1 ≤ m ≤ 2·105) - length of arrays
A and B.

Next line contains m integers a1, a2, ..., am (1 ≤ ai ≤ 109) - array A.

Next line contains m integers b1, b2, ..., bm (1 ≤ bi ≤ 109) - array B.

Output

Output m integers a'1, a'2, ..., a'm - array A' which is permutation of the array A.
```

```
Examples

Input

5
7 3 5 3 4
2 1 3 2 3

Output

4 7 3 5 3

Input

7
4 6 5 8 8 2 6
2 1 2 2 1 1 2

Output

2 6 4 5 8 8 6
```

## Contaminated TACO Test Set Problem

```
Leha like all kinds of strange things. Recently he liked the function F(n, k). Consider
 all possible k-element subsets of the set [1, 2, ..., n]. For subset find minimal
element in it. F(n, k) - mathematical expectation of the minimal element among all k-
element subsets.

But only function does not interest him. He wants to do interesting things with it. Mom
 brought him two arrays A and B, each consists of m integers. For all i, j such that 1
≤ i, j ≤ m the condition A_{i} ≥ B_{j} holds. Help Leha rearrange the numbers in the
array A so that the sum $\sum_{i = 1}^{m} F(A_{i}^{\prime}, B_{i})$ is maximally
possible, where A' is already rearranged array.

-----Input-----

First line of input data contains single integer m (1 ≤ m ≤ 2·10^5) - length of arrays
 A and B.

Next line contains m integers a_1, a_2, ..., a_{m} (1 ≤ a_{i} ≤ 10^9) - array A.

Next line contains m integers b_1, b_2, ..., b_{m} (1 ≤ b_{i} ≤ 10^9) - array B.

-----Output-----

Output m integers a'1, a'_2, ..., a'_{m} - array A' which is permutation of the array A
.

-----Examples-----
Input
5
7 3 5 3 4
2 1 3 2 3

Output
4 7 3 5 3

Input
7
4 6 5 8 8 2 6
2 1 2 2 1 1 2

Output
2 6 4 5 8 8 6
```

# K  UPPER BOUND PERFORMANCE ESTIMATION

Throughout this paper, we regard the CodeContests test set as a black box and use the performance of the whole benchmark as the signal for analyzing different *reasoning*, *instruction*, and *execution feedback*. However, optimizing the performance of these prompt variants on a per-problem level will further boost performance. In this section, we aim to provide an upper bound estimation if we select the CoT prompt based on the oracle, i.e., the best test set performance of each problem in the set of prompts. We do not intend the number presented in this section to be compared with the existing methods presented in the main text, as the performance of the test set is exposed, but rather to provide an estimation of the potential room for improvement.

## K.1  ADAPTIVE CoT PROMPT SELECTION

Based on our grid search of 63 *reasoning* × *instruction* prompts, presented in Appendix C.1 and summarized in Table 3. We post-hoc select the *reasoning* and *instruction* prompts, which induce the highest performance per problem rather than over the whole dataset. Table 12 presents the potential room for single-turn performance improvement on CodeContests test set. The best combination per problem is selected based on the best performance in terms of pass@100, and the pass@1 is reported using the same prompts selected by pass@100.

Table 12: **Upper bound adaptive prompts** on CodeContests test set chosen post-hoc from the 63 prompt single-turn CoT grid search (200 samples per problems generated with temperature 1.0). A combination refers to a *reasoning* × *instruction* prompt. The results for the best combination per dataset are the same as the ones presented in Table 3.

|  | Best combination per dataset | | Best combination per problem | |
| --- | --- | --- | --- | --- |
|  | pass@1 | pass@100 | pass@1 | pass@100 |
| Llama 3.0 8B | 1.5 | 17.3 | 2.5 | 22.6 |
| Llama 3.0 70B | 5.3 | 33.1 | 8.3 | 42.4 |
| Llama 3.1 8B | 4.0 | 26.1 | 5.3 | 41.5 |
| Llama 3.1 70B | 16.1 | 54.1 | 18.3 | 63.1 |

## K.2  ADAPTIVE EXECUTION FEEDBACK GRANULARITY SELECTION

We show in Table 13 the post-hoc selection of *execution feedback* granularity based on Table 8 to estimate the upper bound if we select the best granularity per problem in the multi-turn setting. Since in Table 8, 1@3 is estimated from 20 trajectories generated with temperature 0.2 and 100@300 is estimated from 200 trajectories generated with temperature 1.0, we report the upper bound by selecting the best *execution feedback* granularity separately in both setting.

Table 13: **Upper bound adaptive execution feedback (EF)** on CodeContests test set chosen post-hoc from the 4 execution feedback granularity: binary, failed tests, failed & passed tests, LDB. The number for the best dataset EF is extracted from Table 8. All experiments are in the multi-turn setup with a maximum of 3 turns.

|  | Best dataset EF | | Best problem EF | |
| --- | --- | --- | --- | --- |
|  | 1@3 | 100@300 | 1@3 | 100@300 |
| Llama 3.1 8B | 10.9 | 30.9 | 13.1 | 34.8 |
| Llama 3.1 70B | 29.5 | 56.2 | 33.6 | 58.2 |

