# OpenReview forum: "What Makes Large Language Models Reason in (Multi-Turn) Code Generation?"
_ICLR.cc/2025/Conference — ICLR 2025 Poster_

### Official Review · Reviewer_8tPn · 2024-10-19

**Soundness:** 4
**Presentation:** 3
**Contribution:** 2
**Rating:** 8
**Confidence:** 3

**Summary:**

This paper investigates the effects of various prompting strategies on large language models (LLMs) for code generation tasks, focusing on multi-turn interactions and computational requirements. The authors conduct a comprehensive study using competitive programming benchmarks (CodeContests and TACO) across multiple LLM families and sizes. They explore single-turn vs. multi-turn code generation, chain-of-thought (CoT) prompting techniques, execution feedback in multi-turn settings, and the impact of model size and problem difficulty on prompting effectiveness. The study also examines fine-tuning models on multi-turn CoT data.

**Strengths:**

1.Comprehensive evaluation of CoT prompting strategies across different model sizes and benchmarks. I found the negative results that Reasoning prompts not additive interesting.

2.Usage of pass n@k metric for fairer comparison between single-turn and multi-turn approaches.

3.Thorough analysis of the impact of different error-feedback granularities.

4.Investigation of fine-tuning on multi-turn CoT data and its effects on model behavior, showing effectiveness to internalize CoT steps for LLMs.

**Weaknesses:**

1.Although this work has solid empirical evaluations and good presentations, the novelty is a bit limited by combining existing works together, including CoT strategies, error-feedback granularities, and Pass n@k. The core contribution is primarily a prompt engineering effort combined with the evaluation of existing LLM capabilities in code generation. While valuable, this doesn't represent a significant theoretical or methodological advancement in the field. As for the RFT part, although I think it is a novel method used here, I would like to see more details on it to explain it clearly, with details on how it is finetuned with.

2.Limited theoretical analysis of why certain prompting strategies work better than others. Although the authors presented solid empirical results on the prompting strategies ablation, I will still wonder why it works. Plus, since there is no best CoT for every case, i still think cases can be categorized into a bunch of properties, would it be better to have a task analysis model first, to decide which CoT to be applied based on these properties? I would be happy to see the answer to it.

3.Potential overfitting concerns with fine-tuning on multi-turn CoT data generated from the same benchmark used for evaluation, since currently the training process is not clearly described, i keep this as a doubt and look forward to the authors' response.

4.Figure presentations: I found some figures in the appendix not using vector figures, which are in low-resolution, starting from line 1100 and after. And some figures don't have proper captions, starting from line 1524 and after. The typesetting in the appendix should also be reorganized to avoid large blank pages. In a word, I think the good presentation in the main text should be kept as well in the appendix for a good ICLR paper.

**Questions:**

1.How well do the findings generalize to other types of code generation tasks beyond competitive programming? Are there more complicated benchmarks available to evaluate?

2.How large finetuning dataset is required to train a sufficient model that internalized the CoT? Need more finetuning details.

3.How does the finetuned model perform on unseen code generation tasks?

---

> ### Author Response · Authors · 2024-11-19
> **Response to Reviewer 8tPn (1 / 3)**
>
> We thank the reviewers for the encouraging comments and the time dedicated to this. We address the comments and questions below:
>
> > Although this work has solid empirical evaluations and good presentations, the novelty is a bit limited by combining existing works together, including CoT strategies, error-feedback granularities, and Pass n@k.
>
> We thank the reviewer for the assessment of our work. We now clarify where the paper stand:
>
> Existing works in the code generation domain are substantial, but fragmented, on studying CoT and self-repair (multi-turn) individually. It is still an open question of how different CoT prompts perform when entangled with multi-turn execution feedback, and whether the performance differs at scale. We aim to establish a solid empirical baseline with a systematic experimental survey.
>
> This paper introduces novelties including the insights but also the methodology of our empirical survey in the prompting space (as shown in Figure 3) and at scale (pass@100). We go beyond simple prompt engineering as we identify code generation “trajectories” combining different types of prompts but also external tools (Python code execution). The insights, obtained only from extensive experiments across sampling budget and model sizes, are novel. We are glad that the reviewer finds them interesting. We believe they will motivate further work for fine-tuning LLMs as we started with our RFT experiments.
>
> > As for the RFT part, although I think it is a novel method used here,I would like to see more details on it to explain it clearly, with details on how it is finetuned with.
>
> We thank the reviewer for acknowledging the novelty of our RFT approach on multi-turn CoT traces.
>
> We have enriched **Appendix B** to make sure we include the details for each step in the RFT: generation, filtering and post-processing, deduplication and decontamination. This includes not only the experiment configuration but also the statistics of the data we gathered, such as the total number of tokens. We also add more finetuning details such as all the necessary hyperparameters and design choices.
>
> To explain more clearly here, we use the CoT-retry method (multi-turn CoT) to generate 200 samples per problem on all of the DMC training split. Each sample solution will yield different trajectories either:
>
> - problem statement $\rightarrow$ code 1
> - problem statement $\rightarrow$ code 1 $\rightarrow$ execution feedback + reasoning prompt $\rightarrow$ code 2
> - problem statement $\rightarrow$ code 1 $\rightarrow$ execution feedback + reasoning prompt $\rightarrow$ code 2 $\rightarrow$  execution feedback + instruction prompt $\rightarrow$ code 3
>
> The execution feedback here is based on evaluation on the public tests only. The specific reasoning and instruction prompts are detailed in Appendix G.3. To select which trajectories will be part of the fine-tuning dataset, we evaluate the last code generated (code 1, code 2 or code 3 depending on the number of turns) on the public, private and public tests. We keep trajectories which pass all the tests and discard the rest.
>
> We fine-tune our model on these trajectories by predicting the final code solution based on the context, i.e., problem statement and previous code attempts. The model therefore learns correct code but also how to self-correct based on execution feedback and how to reason based on CoT prompts as highlighted by our results in Section 5.3. I hope this clarifies the RFT approach further.
>
> > Limited theoretical analysis of why certain prompting strategies work better than others. Although the authors presented solid empirical results on the prompting strategies ablation, I will still wonder why it works.
>
> Understanding the mechanism and interpretability of CoT prompting from a theoretical viewpoint is an active research field. For example, Li et al. [1] studied the expressiveness of transformers with CoT under computational complexity framework. Our work is orthogonal to this line of work as the scope of this paper is to establish a comprehensive experimental survey of various CoTs at scale in multi-turn code generation domain with empirical insights.

---

> > ### Author Response · Authors · 2024-11-19
> > **Response to Reviewer 8tPn (2 / 3)**
> >
> > > Plus, since there is no best CoT for every case, i still think cases can be categorized into a bunch of properties, would it be better to have a task analysis model first, to decide which CoT to be applied based on these properties?
> >
> > We take a different approach in the manuscript (Figure 5, with details in Appendix C.2): Instead of categorizing problems, we categorize CoT prompts and find that solution-based reasoning works better than the others. This is an interesting question and resonates with the Q2 of reviewer HALe. We provide an **upper bound performance estimation** here, in which the best CoT prompt for each problem is selected in a **post-hoc** manner:
> >
> > | Model         | Best Combination per Dataset |            | Best Combination per Problem |            |
> > |---------------|------------------------------|------------|-----------------------------|------------|
> > |               | Pass@1                       | Pass@100   | Pass@1                      | Pass@100   |
> > | Llama 3.0 8B  | 1.5                           | 17.3       | 2.5                         | 22.6       |
> > | Llama 3.0 70B | 5.3                           | 33.1       | 8.3                         | 42.4       |
> > | Llama 3.1 8B  | 4.0                           | 26.1       | 5.3                         | 41.5       |
> > | Llama 3.1 70B | 16.1                          | 54.1       | 18.3                        | 63.1       |
> >
> > The results show that there is a large potential to explore better way, for example a trainable module as the reviewer suggests, to decide the CoT taken, and understand the relationship between problem property and the corresponding best CoT. We include these results in our revision **Appendix K.1**, and leave the design of a task analysis model for future work.
> >
> > > Potential overfitting concerns with fine-tuning on multi-turn CoT data generated from the same benchmark used for evaluation, since currently the training process is not clearly described, i keep this as a doubt and look forward to the authors' response.
> >
> > We separate this comments into 2 parts to answer:
> >
> > 1. **Details of training process**
> >
> > We add more details for our RFT data collection and training processes in Appendix B.2 and invite the reviewer to refer to this part. The pipeline covers 3 steps: Generation, Filtering and Post-processing, Deduplication and decontamination. We also detail the statistics and the amount of data in each step in Appendix B.2.1, B.2.2 and B.2.3..The finetuning hyperparameters are detailed in Appendix B.3. Please let us know if the reviewer has additional questions. We are happy to clarify the details.
> >
> > 2. **Overfitting concerns**
> >
> > Our RFT model is trained on CodeContests training set and evaluated on CodeContests valid/test set. We also evaluated this checkpoint on another benchmark the TACO test set to show that it generalizes to other competitive programming benchmarks (see Table 4). For this, we also conduct careful decontamination, with details in the Appendix I.
> >
> > We here conduct additional experiments to evaluate Llama 3.1 70B and the RFT model on the following benchmarks, which the fine-tuning is not directly targeting:
> >
> > **Single-Turn (temperature 0.2). Results reported as pass@1 / pass@10:**
> >
> > | Model          | HumanEval+      | MBPP+          | LiveCodeBench - v4 - Easy | LiveCodeBench - v4 - Medium | LiveCodeBench - v4 - Hard | LiveCodeBench - v4 - All |
> > |----|----|----|-----|----|----|----|
> > | Llama 3.1 70B              | 71.8 / 77.0    | 65.2 / 70.9    | 73.8 / 85.0              | 22.0 / 37.4               | 3.3 / 7.2                | 34.2 / 45.3             |
> > | $\text{Llama 3.1 70B}^{\text{RFT}}$        | 72.1 / 76.9    | 63.5 / 69.2    | 76.2 / 85.7              | 22.0 / 37.0               | 3.5 / 8.0                | 35.1 / 45.3             |
> >
> > **Multi-Turn (temperature 0.2). Results reported as 1@3 / 10@30:**
> > | Model          | LiveCodeBench - v4 - Easy | LiveCodeBench - v4 - Medium | LiveCodeBench - v4 - Hard |
> > |----|----|----|-----|
> > | Llama 3.1 70B              | 82.8 / 94.3              | 30.8 / 49.2                | 4.77 / 9.45              |
> > | $\text{Llama 3.1 70B}^{\text{RFT}}$        | 86.0 / 94.4              | 31.5 / 50.1                | 4.74 / 9.19              |
> >
> > The experiments show that after RFT, the model is not overfitting to CodeContests benchmark. It degrades the performance within 2% on HumanEval+ and MBPP+. It is also the case, while sometimes outperforms the base model, on LiveCodeBench-v4 [2], a benchmark with problems released between May 2023 and Sep 2024 containing 713 problems, under both single-turn and multi-turn setting.
> >
> > We update these experiments in **Appendix B.4**.
> >
> > > Appendix: Figure presentations, type setting, … I think the good presentation in the main text should be kept as well in the appendix for a good ICLR paper.
> >
> > We thank the reviewer for the encouraging comments. We have made the presentation of the Appendix better and more compact in the updated version.

---

> > > ### Author Response · Authors · 2024-11-19
> > > **Response to Reviewer 8tPn (3 / 3)**
> > >
> > > > Q1: How well do the findings generalize to other types of code generation tasks beyond competitive programming? Are there more complicated benchmarks available to evaluate?
> > >
> > > We believe our findings can generalize to other types of code generation tasks however only those with simple code solutions. For the multi-turn setup, we need the full problem context and multiple solutions to fit into an LLM context window. Many instruction prompts also work best with shorter code (such as adding comments after each line), or assume the problems can be solved in a few functions.
> > >
> > > With these constraints in mind, we argue competitive programming benchmarks are the hardest code generation tasks and most interesting to study for CoT as they require the most reasoning. Simpler benchmarks such as HumanEval+ or MBPP+ benefit less from diverse CoT at scale as the model already understands the problem context without extra prompting.
> > >
> > > Beyond competitive programming, we find the recently released BigCodeBench[3] targeting software-engineering-oriented problems interesting, by providing a holistic evaluation of tool use and more complicated instruction following and we are motivated to evaluate it.
> > >
> > > > Q2: How large finetuning dataset is required to train a sufficient model that internalized the CoT? Need more finetuning details.
> > >
> > > We add more details of the dataset size and the statistics (number of tokens, number of multi-turn trajectories) in the Appendix B.2.
> > >
> > > The answer to this question depends on the target domain and the data source: is the objective is to target only a specific benchmark, or general code generation tasks, or more generally, code generation, natural language QA, and math reasoning ability all together? Is the data generated on-policy or by another model? Intuitively, the dataset should grow as the target domain grows to cover the space. And finetuning on on-policy is more data efficient than on off-policy data [4], therefore requiring less data. In our setting, we found that the RFT does not hurt the performance on the other code generation tasks despite the fine-tuning set only containing CodeContests solutions.
> > >
> > > > Q3: How does the finetuned model perform on unseen code generation tasks?
> > >
> > > The finetuned model is trained on CodeContest training set. We conduct careful decontamination, with details in **Appendix I** to enable a contamination-free evaluation on TACO test set and CodeContest test set. To the best of our knowledge, we are not aware of any public efforts on decontaminating these 2 benchmarks.
> > >
> > > For other benchmarks, please see the above tables with experiments on HumanEval+, MBPP+, and LiveCodeBench-v4 to measure the generalization. We evaluate these 3 benchmarks for single-turn tasks. On LiveCodeBench-v4, we also evaluate multi-turn task.
> > >
> > > Our results show that the finetuned model, despite only finetuned on multi-turn trajectories from CodeContests, increases its performance on single-turn LliveCodeBench-v4 Easy, Hard and All, on par on single-turn HumanEval+ and LiveCodeBench-v4-Medium, and slightly degrades the performance on single-turn MBPP+. It also increases the performance on multi-turn LiveCodeBench-v4 Easy and Medium, and has slight performance degradation on Hard split.
> > >
> > > [1] Chain of Thought Empowers Transformers to Solve Inherently Serial Problems. Li et al. https://arxiv.org/pdf/2402.12875
> > >
> > > [2] LiveCodeBench: Holistic and Contamination Free Evaluation of Large Language Models for Code. Jain et al. https://arxiv.org/abs/2403.07974
> > >
> > > [3] BigCodeBench: Benchmarking Code Generation with Diverse Function Calls and Complex Instructions. Zhuo et al. https://arxiv.org/abs/2406.15877
> > >
> > > [4] RL on Incorrect Synthetic Data Scales the Efficiency of LLM Math Reasoning by Eight-Fold. Setlur et al. https://arxiv.org/html/2406.14532v1

---

> > > > ### Author Response · Authors · 2024-11-23
> > > > **Looking forward to further feedback**
> > > >
> > > > Dear Reviewer 8tPn,
> > > >
> > > > Thank you again for the great efforts and the valuable comments. We have carefully addressed the main concerns in detail and updated the manuscript accordingly. We hope you might find the response and additional experiment details (e.g. the finetuning details) we added, satisfactory. We believe that the additional experiments, analysis, and explanations significantly improve the quality of our submission and promote transparency/reproducibility among our community. If our response has addressed some of your concerns, we sincerely hope you will consider raising the score.
> > > >
> > > > We understand that this is a busy period, and reviewing papers requires a lot of time and effort. As the end of the discussion approaches, we are eager to hear more about your comments on our manuscript and our response. We are happy to clarify any further concerns (if any).
> > > >
> > > > Best Regards,
> > > >
> > > > Authors

---

> > > > > ### Comment · Reviewer_8tPn · 2024-11-25
> > > > > **Update with the review**
> > > > >
> > > > > Dear authors:
> > > > >
> > > > > Sorry for the late response. Thank you for your thorough and detailed responses, which have addressed most of my concerns. Although I still think the novelty of this work is a bit limited, it is one of the most sound empirical evaluation on LLM code generation with CoT and feedback techniques, and the usage of RFT technique to show that LLMs can internalize the CoT process is a spotlight of this paper. I have raised my score to reflect the improvement done during the rebuttal period.
> > > > >
> > > > > Best regards,
> > > > >
> > > > > Reviewer 8tPn

---

> > > > > > ### Author Response · Authors · 2024-11-25
> > > > > > **Thank you for your engagement**
> > > > > >
> > > > > > Dear Reviwer 8tPn,
> > > > > >
> > > > > > Thank you very much for your helpful feedback and for acknowledging the improvements made during the discussion period.
> > > > > >
> > > > > > Authors

---

### Official Review · Reviewer_NFbK · 2024-11-04

**Soundness:** 2
**Presentation:** 2
**Contribution:** 2
**Rating:** 5
**Confidence:** 4

**Summary:**

This paper evaluates various prompting strategies in single-turn and multi-turn code generation settings. For single-turn generation, it explores chain-of-thought (CoT) prompting, dividing it into reasoning prompts and instruction prompts. In multi-turn generation, the paper investigates multiple error feedback strategies. The evaluation covers the CodeContests and TACO benchmarks, focusing on Llama models and GPT-4o. The authors also fine-tune the models on multi-turn CoT data to internalize reasoning within the LLMs.

**Strengths:**

-	The paper brings together the evaluation of a wide range of prompting strategies in both single turn and multi-turn for multiple models on a challenging code dataset.
-	Extensive set of experiments with an extensive grid search
-	Interesting idea of splitting the CoT prompts into reasoning, instruction, and feedback so that we can understand what strategies work best for different model sizes and problem difficulties.
-	Multi-turn code generation findings reinforce insights from previous papers --- just multi-turn is not sufficient but combining with CoT might improve performance.
-	Fine-tuning on multi-turn CoT data is a novel approach that shows promising results.

**Weaknesses:**

- **Limited novelty**: There is substantial existing research on CoT and self-repair in the code domain, which somewhat limits the paper's originality.

- COT-Retry prompts seem like an arbitrary combination of prompts. Why this particular combination?
The paper lacks any chain of thought prompting specific to reasoning about why the test failed or how to fix the test.


- For additive reasoning prompts experiment in table 6, what happens if each sample uses a different COT prompt?


Some of the details are missing/not clear that gives me less confidence on the results:

1.	**Baseline for Fine-Tuning**: There should be a baseline that fine-tunes on problem-solution pairs (without CoT data). This would help clarify whether RFT fine-tuning offers benefits beyond simple fine-tuning.

2.	**Clarity on CoT and Multi-Turn CoT**: Tables 1 and 2 could more clearly explain CoT and multi-turn CoT strategies. Are they different across models? Are they also different across sample sizes?

3.	**Another baseline for table 8**: How does table 8 results compare to just resampling from the base model? Is feedback even playing a role in these multi-turn settings?

4.	**Explain Figure 6**: Why does the multi-turn CoT differ from the base model at max_turns=1? Shouldn’t they be identical if CoT-Retry doesn’t apply CoT in the first turn?

5.	**Misinterpretation of results**: Some results are misinterpreted in the text. For instance, Figure 4 uses different axes across charts, making it difficult to interpret trends accurately. . Also, we cannot  say that 1% to 2% is better than 30% to 40% or other way around.

**Questions:**

Please answer the questions in the weakness section.

---

> ### Author Response · Authors · 2024-11-19
> **Response to Reviewer NFbK (1 / 4)**
>
> We thank the reviewer for the constructive review and the time and effort dedicated to understanding our paper and to the review. We address the comments and questions below:
>
> > Limited novelty: There is substantial existing research on CoT and self-repair in the code domain, which somewhat limits the paper's originality.
>
> We thank the reviewer for acknowledging the novelty of our categorization of existing CoT prompts into *reasoning*, *instruction* and *execution feedback* and we are glad that the reviewer finds this interesting.
>
> We acknowledge that there is substantial research on CoT and self-repair in the code generation domain. This breadth of existing work underscores the importance of our study: To the best of our knowledge, existing research focuses either **individually** on CoT or **individually** on self-repair (often using pass@k metrics, not comparable to single turn). Besides, empirical studies on CoT in a systematic way are mostly done in the math reasoning domain, e.g., [1].
>
> With prior work being abundant but fragmented, our study addresses the lack of systematic evaluation and comparison with a large-scale experimental survey of single-turn CoT and multi-turn code generation entangled, and conducts a fairer comparison under pass n@k, as highlighted by Reviewer 8tPn. Our work contributes both methodological clarity and novel insights:
>
> 1.  **Interactions of CoT and self-repair in multi-turn context**, through our new categorization of the prompting space into *reasoning*, *instruction* and *feedback*.
> 2. **Comparison at scale of multi-turn and single-turn with a fixed metric**. To the best of our knowledge, there has not been a large-scale (with 6 models of size ranging from 8B to 405B, up to 200 samples per problem), systematic experimental survey that entangles single-turn CoT and multi-turn CodeGen under a consistent evaluation metric, such as pass n@k. We adopted this metric to ensure fairer comparisons, as highlighted by Reviewer 8tPn.
> 3. **Novel RFT method** on multi-turn CoT data to internalize both reasoning abilities and capacity for the model to self-correct between turns which further boosts performance.
>
> Beyond novelty in the methods studied, we also provide key insights (3 from single turns, 3 from multi turns) from our extensive experiments at scale worth sharing with the community.
>
> > COT-Retry prompts seem like an arbitrary combination of prompts. Why this particular combination? The paper lacks any chain of thought prompting specific to reasoning about why the test failed or how to fix the test.
>
> We separate this comment into 2 parts to answer:
> 1. **Regarding the CoT-retry**:
>
> Thank you for pointing this out, we have detailed this further in Appendix G.3 (updated version). We use this combination as our experiments show that it’s the most generalizable across models and sample sizes.  For all reasoning and instruction prompts combinations on the CodeContest test set, we calculated the delta of pass rate improvement with respect to the baseline, i.e., $\text{delta} = (x - \text{baseline})/ \text{std}(x)$ where the baseline is the pass rate without any CoT prompt, $x$ is the pass rate for a combination with a specific model. We average this delta across all models (Llama 3.0 series, Llama 3.1 series and GPT-4o) and look at the **Top 3** best performing combinations for each sample size (i.e. $k$ in pass@$k$). We empirically tested the combinations that appear the most often in the Top 3 and the “code solution” reasoning with “weak solution” instruction performed the best the DMC and TACO datasets.
>
> 2. **Regarding the prompt to reason and fix test failure**:
>
> We did not include a specific prompt to reason about why the test failed as we found that capable models, such as Llama 3.1 series, already exhibit basic ability to analyze the failing public tests and fix them without explicitly prompting to do so. With our CoT-retry, the first turn is the model’s first attempt to a solution then second turn the model adapts the first solution to edge cases to make sure it generalizes, and finally in the third turn we induce diversity by making the model give up on its current attempt and try a naive solution instead.

---

> ### Author Response · Authors · 2024-11-19
> **Response to Reviewer NFbK (2 / 4)**
>
> We conduct experiments in which we use explicit prompting on this (namely the “Fixme” and “Analyze” and the combination) after giving the execution feedback. We show the prompts here:
>
> - **Retry (default)**: “Give it another try.” (Used in the paper)
> - **Fixme**: “Generate a fixed version of the program to fix the failing test.”
> - **Analyze $\rightarrow$ Retry**: “Analyze the execution feedback. If runtime exception, identify the source. If wrong answer, simulate and analyze how the input maps to the actual output in your code and where it differs from the expected output. After that, give it another try.”
> - **Analyze $\rightarrow$ Fixme**: “Analyze the execution feedback. If runtime exception, identify the source. If wrong answer, simulate and analyze how the input maps to the actual output in your code and where it differs from the expected output. After that, generate a fixed version of the program to fix the failing test.”
>
> We show the multi-turn results (reported as 1@3 / 100@300) on CodeContests test set below:
> | Model         | Retry | Fixme | Analyze$\rightarrow$Retry | Analyze$\rightarrow$Fixme |
> |---------------|-----------------|--------------------|-----------------------------|---------------------|
> | Llama 3.1 8B  | 7.0 / 30.4      | 6.7 / 29.3 | 6.6 / 30.0         | 6.3 / 27.5         |
> | Llama 3.1 70B | 24.1 / 56.2     | 25.2 / 55.7 | 25.2 / 54.6        | 24.9 / 55.9        |
>
> As shown in the table. explicitly prompting the model to focus on the failing tests and fix it degrades the performance for the 8B model in 1@3 and 100@300. For the 70B model, the 1@3 increases by 1.1% while the 100@300 drops. This shows an interesting exploration-exploitation trade-off. As we posit in the Appendix C.6, the bottleneck of solving competitive programming problems like CodeContests is rather the algorithmic reasoning ability but not the bug-fixing ability.
>
> We have added this experiment and the analysis in **Appendix E.1**.
>
> We here share another insight here from what we observe in the experiments: Explicit prompting to fix the failing tests might work in a fully-observable environment (i.e., all the tests are used for feedback), but might encounter challenges in a partially-observable environment (i.e., only public tests as feedback and private tests as evaluation), which is our setting.
>
> > For additive reasoning prompts experiment in table 6, what happens if each sample uses a different COT prompt?
>
> To clarify, in table 6 we tried to understand whether adding reasoning and instruction prompts would further boost performance; however, we keep the same prompt combination for all samples. We pick the best performing CoT candidates from the single-turn 8B setup; +1 reasoning we add a second turn of reasoning before code generation and +1 instruction we add an instruction prompt at the second turn of multi-turn. We clarify this further in the Appendix Table 7.
>
> Obtaining the Oracle best number would require 9x7x9x7 prompt combinations multiplied by the number of samples. Our results for table 6 are obtained from a smaller grid search where we select CoT prompt candidates that perform well on a smaller scale, e.g., single-turn setting or 8B model.
>
> However, for single-turn, 1 reasoning x 1 instruction, we provide the **upper bound estimation** (obtained from a grid search of 63 runs) of choosing the best reasoning and instruction prompts per sample in a **post-hoc** manner. This increases the performance significantly while the prompts would be heavily cherry-picked per sample and the test set performance is exposed.
>
> | Model         | Best Combination per Dataset |            | Best Combination per Problem |            |
> |---------------|------------------------------|------------|-----------------------------|------------|
> |               | Pass@1                       | Pass@100   | Pass@1                      | Pass@100   |
> | Llama 3.0 8B  | 1.5                           | 17.3       | 2.5                         | 22.6       |
> | Llama 3.0 70B | 5.3                           | 33.1       | 8.3                         | 42.4       |
> | Llama 3.1 8B  | 4.0                           | 26.1       | 5.3                         | 41.5       |
> | Llama 3.1 70B | 16.1                          | 54.1       | 18.3                        | 63.1       |
>
> We update this in **Appendix K.1**.

---

> > ### Author Response · Authors · 2024-11-19
> > **Response to Reviewer NFbK (3 / 4)**
> >
> > > There should be a baseline that fine-tunes on problem-solution pairs (without CoT data). This would help clarify whether RFT fine-tuning offers benefits beyond simple fine-tuning.
> >
> > Thank you for this very constructive suggestions. We include the ablation of whether including CoT response below and report the performance (1@3 to 100@300) on multi-turn CodeContests below and have updated it with details of each experiment run in **Appendix E.3**. These results are comparable to the results in Table 1.
> >
> > In the ablation, we remove the CoT response in the Llama 3.1 70B generated data. We also include an SFT baseline that trains on CodeContests training set following the data cleaning in [2].
> >
> > | Code Source | ST  | MT  | CoT Response | 1@3  | 10@30 | 100@300 |
> > |----|----|----|----|----|----|----|
> > | Llama 3.1 70B (Base Model)             | X          | X   | X              | 24.1  | 43.8   | 56.2    |
> > | CodeContests/train (SFT)           | $\checkmark$          | X   | X             | 16.6  | 33.6   | 44.9    |
> > | Llama 3.1 70B (RFT)       | $\checkmark$          | X   | X             | 26.8  | 47.5   | 58.3    |
> > | Llama 3.1 70B (RFT)       | $\checkmark$          | $\checkmark$   | X            | 28.9  | 49.2   | 60.1    |
> > | Llama 3.1 70B (RFT)       | X          | $\checkmark$   | X             | 29.1  | 50.1   | 60.0    |
> > | Llama 3.1 70B (RFT)       | $\checkmark$          | $\checkmark$   | $\checkmark$            | 29.1  | 49.6   | 60.0    |
> > | Llama 3.1 70B (RFT)       | X          | $\checkmark$   | $\checkmark$   | 29.7  | 50.5   | 61.1    |
> >
> > We observe that SFT on the given training set degrades the performance. RFT on the self-generated data provides the performance boost, on top of which including CoT response further increases the performance. We put ablation analysis in **Appendix E.3**.
> >
> > > Tables 1 and 2 could more clearly explain CoT and multi-turn CoT strategies. Are they different across models? Are they also different across sample sizes?
> >
> > They are different across models as we look at the maximum performance of each prompt. They are the same across sample sizes. We estimate the n@k from the same set of samples. We make this clear in the revision.
> >
> > > How does table 8 results compare to just resampling from the base model? Is feedback even playing a role in these multi-turn settings?
> >
> > Interesting question. This is one of the motivations that drives the paper, in which we aim to compare single-turn (“just resampling from the base model”) and multi-turn in a fairer way as emphasized in section 2.2 “limitations of pass@k”. We add a row in Table 8 below to compare with its equal-compute single-turn counterpart. In general throughout our paper, single-turn experiments with k > 1 correspond to ressampling from the base model.
> >
> > We update the experiment's result here and in the Appendix D.2 (updated version).
> >
> > |            | Llama 3.1 70B - 1@3 | Llama 3.1 70B - 100@300 | Llama 3.1 8B - 1@3 | Llama 3.1 8B - 100@300 |
> > |---------------------------|-------------------|-----------------------|------------------|----------------------|
> > | Repeated Sampling         | 27.3             | 53.5                 | 11.9            | 28.0                |
> > | Binary Feedback           | 28.8             | 55.9                 | 10.9            | 30.9                |
> > | Failed Tests Feedback (default) | 29.5       | 56.2                 | 10.9            | 29.5                |
> >
> > > Is feedback even playing a role in these multi-turn settings
> >
> > This is an interesting question. From the above table, compared to single-turn repeated sampling, adding feedback does increase performance in some settings. For Llama 3.1 70B, 1@3 increases by 2.2%, and 100@300 increases by 2.7%.
> >
> > Although there are some settings where we do not observe performance gains using execution feedback, our analysis helps to understand why: **our key insight about the role of feedback is that they bias the model’s multi-turn behavior into a more exploratory one or a more exploitative one (Sec 5.2 and Figure 6)**. This possibly explains why providing fine-grained debugger information (LDB [2]) works well on relatively easy benchmarks such as HumanEval but not on CodeContests, as the former could be bottlenecked by the accidental bug during code generation while the latter is bottlenecked by model’s algorithmic reasoning ability. This is an insight that could benefit the community from controlling the model's multi-turn behavior to adapt to their specific use case.
> >
> > Also, we provide an upper bound performance if we select the best feedback granularity per problem. This result is updated in **Appendix K.2**.
> >
> > | Model         | Metric               | 1@3  | 100@300 |
> > |---------------|----------------------|-------|---------|
> > | Llama 3.1 70B | Best Fixed EF        | 29.5  | 56.2    |
> > | Llama 3.1 70B | Best EF Per-Problem  | 33.6  | 58.2    |
> > | Llama 3.1 8B  | Best Fixed EF        | 10.9  | 30.9    |
> > | Llama 3.1 8B  | Best EF Per-Problem  | 13.1  | 34.8    |

---

> > > ### Author Response · Authors · 2024-11-19
> > > **Response to Reviewer NFbK (4 / 4)**
> > >
> > > > Explain Figure 6: Why does the multi-turn CoT differ from the base model at max_turns=1? Shouldn’t they be identical if CoT-Retry doesn’t apply CoT in the first turn?
> > >
> > > This is an error we made: we report the performance for the base model in Figure 6 using the eval number from an older codebase where there was a bug for dynamic RoPE scaling (a feature newly introduced for Llama 3.1 series). This impacts the Figure 6 and Figure 2 for the base model performance, which makes the pass rate lower.
> > >
> > > To make sure other experiments weren’t impacted, we checked all our code snapshot for experiments and confirmed that the other results reported are from the bug-free codebase. We fix Figure 2 and 6 (Figure 6 being Figure 7 in the revised version). The fixed plots reinforce the conclusion of each, namely: (1) Simply scaling the number of turns is not compute optimal. (2) CoT-Retry increases the multi-turn performance and RFT lifts the whole curve upward.
> > >
> > > We thank the reviewer for this careful review and making our manuscript better.
> > >
> > > >  Some results are misinterpreted in the text. For instance, Figure 4 uses different axes across charts, making it difficult to interpret trends accurately. Also, we cannot say that 1% to 2% is better than 30% to 40% or other way around.
> > >
> > > Our reasoning around not sharing y axis is because we do not compare pass rates across difficulty splits rather look at different prompts within a split. We think it’s reasonable to compare relative improvements under this context specifically the percentage increase of the best CoT with respect to no CoT. We have made it clearer in the revision. Instead of comparing 2% to 4% and 28% to 32%, we will make it explicit that we compare the relative percentage increase.
> > >
> > > [1] Towards Understanding Chain-of-Thought Prompting: An Empirical Study of What Matters. Wang et al. https://arxiv.org/pdf/2212.10001
> > >
> > > [2] LLM-Assisted Code Cleaning For Training Accurate Code Generators. Jain et al. https://arxiv.org/abs/2311.14904

---

> > > > ### Author Response · Authors · 2024-11-23
> > > > **Looking forward to further feedback**
> > > >
> > > > Dear Reviewer NFbK,
> > > >
> > > > Thank you again for the great efforts and the valuable comments. We have carefully addressed the main concerns in detail and updated the manuscript accordingly. We hope you might find the response satisfactory. We believe that the additional experiments, analysis, and explanations significantly improve the quality of our submission. If our response has addressed some of your concerns, we sincerely hope you will consider raising the score.
> > > >
> > > > We understand that this is a busy period, and reviewing papers requires a lot of time and effort. As the end of the discussion approaches, we are eager to hear more about your comments on our manuscript and our response. We are happy to clarify any further concerns (if any).
> > > >
> > > > Best Regards,
> > > >
> > > > Authors

---

> > > > > ### Comment · Reviewer_NFbK · 2024-11-26
> > > > > **Reply to author**
> > > > >
> > > > > Thank you for your rebuttal and the new experiments. These new experiments significantly enhance the paper's value and have strengthened my confidence in its contributions. Accordingly, I have increased my score.
> > > > >
> > > > > However, I also feel that the paper is not doing a full justice to either fully exploring the COT landscape or fully exploring the multi-turn landscape. There are already way too many experiments in the paper and the rebuttal and the results could be very confusing for a reader. And still there are several other aspects to be explored (as mentioned by other reviewers as well, such as having a task analysis model, using multiple COT prompts for different samples, etc.). Actually, it would have been better to split the work into two separate papers since COT and multi-turn are mostly orthogonal aspects.
> > > > >
> > > > > I would also highly encourage the authors to refine the writing further, find a better way to organize the results in the paper, and explicitly highlight aspects of the problems still unexplored in this paper.

---

> > > > > > ### Author Response · Authors · 2024-12-02
> > > > > > **Thank you for your engagement**
> > > > > >
> > > > > > We thank the reviewer for the constructive feedback and for acknowledging the value added by our new experiments and the improvement made. We appreciate the increased score and are glad that our revisions have strengthened the reviewer's confidence in our contributions.
> > > > > >
> > > > > > **Our motivation**: We agree that these are broad topics with many aspects worth investigating. Also, exactly because of this, we believe that our paper provides a crucial first step towards understanding the interactions between CoT (tokens generated by the model internally) and self-repair/multi-turn (tokens from the external environment) in the code generation domain, which has been under-explored in existing research.
> > > > > >
> > > > > > > Splitting the paper into 2 parts focusing on the CoT and multi-turn for orthogonal aspects
> > > > > >
> > > > > > We would like to share our vision and argue that it is important to have a cohesive presentation of key insights/empirical findings when these two methods coupled together:
> > > > > >
> > > > > > If we prob beyond self-repair/multi-turn, which we see as the simplest *agentic* setting where model could interact with coding environment and gather feedback, it would be code agent that navigates and interacts in a more complex environment (e.g. environment being a repository, create code file, inspect file content) and more turns of feedback from runtime are needed (e.g. getting output from the common linux command (`ls`, `cat` etc), file running output, unit test checking, CI testing). Therefore, the model ability in reasoning (e.g. using CoT) about these runtime feedback is essential if we want to make further progress and we believe our work laid a foundation for pushing along this direction.
> > > > > >
> > > > > > > still there are several other aspects to be explored ... Better presentation ...
> > > > > >
> > > > > > We agree that more sophisticated methods designs, as mentioned by the reviewers, are worth-exploring. We regard what we present in this manuscript as a solid empirical foundation for these further explorations / potential extensions of our research. For example, we have provided the empirical upper bound estimation for adaptive prompt/execution-feedback selection in Appendix K. We explicitly highlight the limitations of our study and the aspects of the problem that remain unexplored, to encourage future research in this direction for the community.
> > > > > >
> > > > > > In the current revision, we extract our key insights and major experimental findings in the main text, with ablations and detailed settings in the corresponding Appendix. For a better presentation, we will further refine the writing.
> > > > > >
> > > > > >
> > > > > > Thank you again for your time and efforts in the review and engaging with the discussion.

---

### Official Review · Reviewer_HALe · 2024-11-04

**Soundness:** 4
**Presentation:** 4
**Contribution:** 3
**Rating:** 8
**Confidence:** 3

**Summary:**

The paper investigates the effects of varied prompting techniques, including chain-of-thought (CoT), reasoning, and execution feedback prompts, on the performance of large language models (LLMs) in multi-turn code generation. It focuses on understanding the mechanics and efficacy of these prompts on complex programming tasks using benchmarks like CodeContests and TACO across models of different sizes (e.g., Llama 3.0/3.1, GPT-4o). Unlike simpler code generation tasks, competitive coding demands algorithmic reasoning and has not been widely explored with unified evaluation metrics for multi-turn code generation.

Key contributions include:

1. **Framework for Multi-Turn Code Generation**: The paper establishes a unified framework that enables a mix of single-turn and multi-turn prompting techniques, incorporating reasoning, instruction, and feedback prompts. This approach aims to build robust baselines for multi-turn performance and to measure trade-offs between sampling budgets and model capabilities.

2. **Experimental Insights**: The authors’ extensive grid search reveals that combining reasoning and instruction prompts significantly enhances single-turn model performance, especially for larger models and more difficult tasks. However, for multi-turn settings, while combining with CoT prompts boosts performance, using multi-turn alone has limited benefits and can even reduce performance at higher sample budgets due to less diverse outputs.

3. **Internalized CoT**: The study adapts the RFT, which allows models to internalize multi-turn CoT reasoning during training. RFT-trained models demonstrate improved performance and reasoning abilities in the absence of CoT prompts at inference, showing promise for scaling up multi-turn code generation capabilities.

4. **Exploration-Exploitation Dynamics**: The analysis indicates that detailed feedback can induce exploitative behavior in models, which is beneficial for simpler problems but counterproductive for complex ones where diversity in code generation is critical. This highlights the need for feedback strategies that balance between exploiting known solutions and exploring new ones, particularly in high-difficulty settings.

In conclusion, this work underscores that multi-turn code generation is not inherently beneficial without adaptive strategies like CoT prompts and RFT. The findings advocate for tailored prompting configurations based on problem difficulty and suggest that further advancements in multi-turn code generation could benefit real-world applications such as repository-level LLM agents capable of reasoning, self-repair, and complex interaction with environments.

**Strengths:**

**Originality**:
This work provides a unique contribution by systematically exploring multi-turn code generation for large language models (LLMs) through diverse prompt configurations, including **chain-of-thought (CoT)**, **reasoning prompts**, and **execution feedback**. While prior work has explored CoT and single-turn code generation, this paper’s emphasis on **multi-turn prompting in competitive programming benchmarks** addresses an under-explored area. The study’s integration of multi-turn CoT with finely tuned execution feedback as a **comprehensive prompting framework** offers an original approach to improve model reasoning in multi-turn contexts. Additionally, the use of **Rejection Sampling Fine-tuning (RFT)** to internalize CoT reasoning across turns is a creative and impactful extension, enabling models to generalize reasoning capabilities without prompting at inference time.

**Quality**:
The paper demonstrates high quality in its rigorous experimental design and breadth of evaluations. Through an extensive **grid search** of prompt combinations, the authors provide a robust dataset for comparing single-turn and multi-turn settings across a **variety of model architectures and sizes** (Llama 3.0/3.1, GPT-4o). The benchmarks, CodeContests and TACO, are well-chosen for evaluating competitive coding capabilities, allowing the paper to contribute valuable insights for LLM applications in algorithmic and complex reasoning tasks. Methodologically, the work includes detailed ablations, sensitivity analyses, and cross-model comparisons, ensuring that the findings are well-supported and generalizable.

**Clarity**:
The paper is well-structured, clearly outlining the motivation, objectives, and experimental setup. The **framework for prompt deconstruction** is presented systematically, with visual aids like **Figure 1** that clarify the operational structure of single- and multi-turn code generation. Each type of prompt (reasoning, instruction, feedback) is explained with appropriate examples, making the framework comprehensible for readers. The authors’ discussion on **exploration-exploitation dynamics** and performance limitations in multi-turn settings enhances clarity by contextualizing how feedback granularity impacts model behavior. Additionally, the conclusions are tied back to the primary research questions, reinforcing the relevance of the findings.

**Significance**:
The study makes significant strides in understanding and advancing multi-turn reasoning for code generation in LLMs. By introducing a unified prompting framework and evaluating its efficacy in algorithmically demanding benchmarks, the paper sets a strong foundation for future research in **multi-turn, reasoning-heavy applications** for LLMs. The RFT approach has implications beyond competitive programming, as it suggests that **prompted reasoning behaviors can be internalized** within the model, potentially reducing computational costs for other complex, multi-step tasks. Given the trend toward **LLM-based agents** capable of autonomous decision-making, this paper’s insights into adaptive CoT, feedback prompting, and exploration-exploitation trade-offs are timely and have high applicability.

**Conclusion**:
In summary, this paper’s originality in multi-turn prompting, its methodological rigor, clarity, and significance make it a valuable contribution to the field of LLM code generation. The findings not only enhance understanding of CoT in multi-turn settings but also provide practical strategies for optimizing reasoning in LLMs, making this work highly relevant for both researchers and practitioners.

**Weaknesses:**

While the paper makes significant contributions to the understanding of multi-turn code generation in LLMs, there are a few areas where improvements could be made to strengthen the study further.

**1. Limited Scope of Multi-Turn Framework**
The paper’s framework, while well-designed for competitive programming benchmarks, focuses primarily on **chain-style multi-turn code generation**. This approach limits the exploration of more complex **tree-structured trajectories** or **backtracking mechanisms** that could better emulate real-world multi-turn interactions in agent-based LLM applications. Exploring additional trajectory structures—such as decision-tree-style paths where LLMs could backtrack or revise earlier steps—would add depth to the study and provide insights into more adaptive reasoning scenarios. Extending the framework to incorporate these structures could lead to a more comprehensive understanding of multi-turn reasoning dynamics.

**2. Potential Bias in Rejection Sampling Fine-Tuning (RFT)**
Although Rejection Sampling Fine-Tuning (RFT) is a compelling method for embedding reasoning behavior, the paper’s reliance on **correctly generated trajectories** may introduce a bias toward successful patterns, potentially limiting the model’s adaptability to a broader range of problem-solving approaches. By focusing only on “successful” reasoning patterns, the model may overfit to specific prompt types or problem formats. Future work could consider **diversifying the fine-tuning data** to include not only correct but also nearly correct attempts with varied prompts, allowing models to learn from a broader array of problem-solving paths.

**3. Dependence on High-Resource Settings**
The paper’s approach requires considerable **computational resources** due to the grid search over a large set of prompt combinations and the use of multi-turn, CoT, and feedback prompts across different model sizes. This may limit the practical application of the findings for research groups or practitioners with restricted compute budgets. Introducing methods for **adaptive prompt selection** that can identify effective prompt configurations more efficiently (e.g., meta-learning approaches) could make the approach more accessible and scalable, especially for real-world applications where computational resources may be constrained.

**4. Limited Discussion of Prompt Failure Modes**
The paper could benefit from a more detailed **analysis of prompt failure modes**, particularly in cases where certain prompt combinations degrade performance in multi-turn settings. While the paper briefly notes that CoT may harm performance by obstructing LLM context or encouraging suboptimal plans, it lacks a thorough exploration of specific cases where multi-turn CoT and feedback combinations lead to failure. A more in-depth examination of these failure patterns, especially for challenging tasks, would provide actionable insights into optimizing prompt configurations and mitigating these issues in future iterations.

**Conclusion**
In summary, while the paper delivers strong contributions to multi-turn prompting strategies, expanding the framework’s scope to include more diverse trajectory structures, and refining RFT to include a wider range of training data. Additionally, addressing the high computational costs and analyzing prompt failure modes more deeply would make the work more robust and accessible for practical application. These improvements could further strengthen the study and support its stated goal of advancing multi-turn reasoning capabilities in LLMs.

**Questions:**

1. Diversity in Rejection Sampling Fine-Tuning: The paper fine-tunes the model on correctly generated trajectories. Given the potential for this approach to bias the model towards successful solutions, would the authors consider including near-correct or diverse trajectories to introduce a broader range of reasoning paths? This could enhance the model’s adaptability to different problem-solving strategies.
2. Adaptive Prompt Selection for Reducing Computational Requirements: The extensive grid search across prompt types and model configurations is resource-intensive. Have the authors explored adaptive or meta-learning techniques to streamline prompt selection, reducing computational demands while still achieving optimal prompt configurations? Such techniques could make the framework more accessible for settings with limited computational resources.
3. Detailed Analysis of Prompt Failure Modes: The paper mentions that certain CoT prompt configurations may degrade performance, particularly in multi-turn settings with higher sample budgets. Could the authors provide a more granular analysis of these failure modes? Understanding specific prompt combinations or scenarios where performance degrades could guide practitioners in avoiding ineffective configurations.
4. Balancing Exploration and Exploitation in Feedback Prompts: The study notes that finer-grained feedback prompts may encourage exploitative behavior in models, potentially limiting performance on complex tasks. Have the authors considered any strategies for adjusting the feedback granularity dynamically based on task difficulty or performance? This could help achieve a balance between exploration and exploitation for more challenging problems.
5. Transferability of the Findings to Other LLM Architectures: The paper experiments primarily with the Llama series and GPT-4o models. To what extent do the authors believe that the findings, especially related to prompt configurations and RFT, would transfer to other LLM architectures with different pretraining characteristics? For instance, would models with fewer parameters or more domain-specific training exhibit similar responses to the same prompting strategies?
6. Future Directions for Enhancing Multi-Turn Reasoning in LLMs: Given the current results, what specific directions do the authors see as most promising for advancing multi-turn reasoning capabilities? Are there any particular technical approaches, benchmarks, or modifications to current LLM architectures that the authors believe could improve multi-turn performance on reasoning-heavy tasks?

---

> ### Author Response · Authors · 2024-11-19
> **Response to Reviewer HALe (1 / 3)**
>
> We thank the reviewers for the detailed, encouraging and thought-provoking review and the time dedicated to this. We here address the comments and questions below:
>
> > Q1: Diversity in Rejection Sampling Fine-Tuning: The paper fine-tunes the model on correctly generated trajectories. Given the potential for this approach to bias the model towards successful solutions, would the authors consider including near-correct or diverse trajectories to introduce a broader range of reasoning paths? This could enhance the model’s adaptability to different problem-solving strategies.
>
> Including near-correct trajectories for fine-tuning as well as more complex loss functions could be interesting extensions of our work and has been explored with other reasoning tasks. Our efforts for encouraging diverse trajectories include:
> - We use a high temperature 1.0 for sampling.
> - Amongst the correct trajectories, we conduct finetuning on multi-turn trajectories, in which diverse incorrect solutions in the previous turns are also included.
> - We conduct downsampling using LSH-based deduplication to encourage diverse samples being kept, with details in Appendix B.2 (updated version).
>
> We did not include incorrect trajectories in this paper as we focus on understanding inference techniques and view our RFT method as a way to explore the removal of the need for explicit prompting. Simple finetuning on correct trajectories with the LLama 3.1 70B already displayed the key results we wanted: performance boost in contrast with inference-only CoT (Table 1), better self-repair capacities (Figure 8 and 9), and generalization to other benchmarks (Table 4).
>
> A way we think of including near-correct trajectories is to encourage correct reasoning even though final solutions are incorrect. For filtering these trajectories, a Process-Reward Model (PRM) or a “critic” LLM could be used to rate the reasoning correctness of CoT, in order to promote “progress” besides “correctness”. This is an interesting research direction; we added it as a suggestion in section 7 (Limitation) of our paper.
>
> > Q2: Adaptive Prompt Selection for Reducing Computational Requirements: The extensive grid search across prompt types and model configurations is resource-intensive. Have the authors explored adaptive or meta-learning techniques to streamline prompt selection, reducing computational demands while still achieving optimal prompt configurations? Such techniques could make the framework more accessible for settings with limited computational resources.
>
> We agree that the extensive grid search, especially at scale presented in this paper, is computationally expensive. However it yields insights regarding a wide range of models in the fields of code generations. These insights should help the community identify promising reasoning and instruction prompts, number of turns to use for multi-turn and feedback granularity in their experiments. We also emphasize the pass n@k metric rather than pass@k to take into account the computational budget with all our experiments. Also, several insights can transfer from one experimental setting to another. For example, the optimal prompt strategy generalizes from one model to another, or from one benchmark to the other:
>
> - In Appendix C.3, we show that the best CoT, i.e. reasoning and instruction prompt combination, found with Llama 3.0 8B on TACO could be directly ported to Llama 3.1 8B and 70B models.
> - In Table 4, we show that the best multi-turn CoT found on CodeContests can be directly used on TACO test set. We also evaluated the RFTed model trained on CodeContest training set on TACO test set.
>
> A model to select the best prompt can eventually reduce the computational cost in inference time, while training such a model could be expensive. As the reviewer mentioned, adaptive prompt selection could be done either through black-box optimization, such as bayesian optimization, or through optimizing the prompt directly by gradient, such as the line of work of prompt tuning. We cover this discussion in section 7 (Limitation) of the revision.
>
> However, we provide an **upper bound estimation** of the performance for the adaptive prompt selection from our grid search. This is done by a **post-hoc** selection of the CoT per problem by the best performance it induces. We here show the single-turn performance on CodeContests test set under this setting. We do not intend the number presented here to be compared with existing numbers as the test set performance is exposed, and therefore should be treated as an upper bound estimation to show how large the room for improvement.

---

> > ### Author Response · Authors · 2024-11-19
> > **Response to Reviewer HALe (2 / 3)**
> >
> > | Model         | Best Combination per Dataset|            | Best Combination per Problem |            |
> > |---------------|------------------------------|------------|-----------------------------|------------|
> > |               | Pass@1                       | Pass@100   | Pass@1                      | Pass@100   |
> > | Llama 3.0 8B  | 1.5                           | 17.3       | 2.5                         | 22.6       |
> > | Llama 3.0 70B | 5.3                           | 33.1       | 8.3                         | 42.4       |
> > | Llama 3.1 8B  | 4.0                           | 26.1       | 5.3                         | 41.5       |
> > | Llama 3.1 70B | 16.1                          | 54.1       | 18.3                        | 63.1       |
> >
> > We update the manuscript to include this Table in Appendix K.1.
> >
> > > Q3: Detailed Analysis of Prompt Failure Modes: The paper mentions that certain CoT prompt configurations may degrade performance, particularly in multi-turn settings with higher sample budgets. Could the authors provide a more granular analysis of these failure modes?
> >
> > Thank you for pointing this out, we highlight examples of failures of CoT in Appendix K with 2 examples. We observed that on simple problems the model might focus too much on answering the instruction prompt rather than solving the problem which can lead to brute force solutions or repetitive code with the “weak solution” prompt. We can include more examples with other prompts, specifically reasoning prompts. For example, “generate intermediate variables” often makes the model generate many unhelpful variables which overcomplexifies its final code solution (Example 3 in Appendix K).
> >
> > > Q4: Balancing Exploration and Exploitation in Feedback Prompts: The study notes that finer-grained feedback prompts may encourage exploitative behavior in models, potentially limiting performance on complex tasks. Have the authors considered any strategies for adjusting the feedback granularity dynamically based on task difficulty or performance?
> >
> > We do not explore additional components to adaptively adjust which execution feedback (EF) granularity is chosen to put into the LLM context. Future work could benefit from the same techniques for adaptive prompt selection, such as introducing a trainable module, to select the EF granularity according to the given problem statement or difficulty.
> >
> > We provide an upper bound estimation of the performance if there is an oracle balance optimally, in a post-hoc manner, the EF granularity. This is done by selecting the best EF per problem by the best performance it induces.
> >
> > | Model         | Metric               | 1@3  | 100@300 |
> > |---------------|----------------------|-------|---------|
> > | Llama 3.1 70B | Best Fixed EF        | 29.5  | 56.2    |
> > | Llama 3.1 70B | Best EF Per-Problem  | 33.6  | 58.2    |
> > | Llama 3.1 8B  | Best Fixed EF        | 10.9  | 30.9    |
> > | Llama 3.1 8B  | Best EF Per-Problem  | 13.1  | 34.8    |
> >
> > We update the manuscript to include this Table in Appendix K.2.

---

> ### Author Response · Authors · 2024-11-19
> **Response to Reviewer HALe (3 / 3)**
>
> > Q5: Transferability of the Findings to Other LLM Architectures: The paper experiments primarily with the Llama series and GPT-4o models. To what extent do the authors believe that the findings, especially related to prompt configurations and RFT, would transfer to other LLM architectures with different pretraining characteristics? For instance, would models with fewer parameters or more domain-specific training exhibit similar responses to the same prompting strategies?
>
> We experimented with 3 different model series (llama 3.0, llama 3.1 and gpt4o) as well as 6 different architectures to make sure our results would be generalizable. This diversity was chosen to ensure that our findings are robust and not limited to a single model. Based on this experimental breadth, we are confident that the observed patterns in prompt configurations and RFT applicability hold across architectures with similar design principles.
>
> However, we acknowledge that certain characteristics, such as pretraining data distribution, scale, and domain-specificity, can influence a model’s response to prompting strategies. More empirical evidence is needed to investigate the influence of these factors, especially data distribution, on model’s behavior / performance.
>
> To address the reviewer’s specific point, while we believe that our findings are generalizable within the family of large, general-purpose LLMs, the performance of smaller or more specialized models using our proposed strategies warrants further investigation. We plan to explore these dimensions in future work and encourage the community to extend these findings to different model types.
>
> > Q6: Given the current results, what specific directions do the authors see as most promising for advancing multi-turn reasoning capabilities? Are there any particular technical approaches, benchmarks, or modifications to current LLM architectures that the authors believe could improve multi-turn performance on reasoning-heavy tasks?
>
> We acknowledge that current LLMs are trained on code infilling loss, therefore they are capable of generating code blocks of coherent code, but training them to reason about code execution and to understand execution feedback remains a challenge.  We believe that better understanding of **execution feedback** is necessary for LLMs to benefit the most from the multi-turn setup. This could be improved by training a specialized code world model.
>
> Also, as the reviewer points out, adaptive prompt selection could be a promising path forward given the encouraging upper pound we give in the previous response.
>
> Although out of scope for this paper, we could see agentic code generation such as SWE-Bench as a more challenging multi-turn setup and an interesting benchmark for evaluating reasoning. In this paper, we focus on an empirical survey to maximize current LLM performance with a fixed budget at inference. We hope the limits we find with LLM reasoning can motivate further work on bridging the gap.

---

### Author Response · Authors · 2024-11-19

We thank all the reviewers for providing constructive review of our paper. We appreciate the time spent to analyze our paper specifically reading through the appendix section as well as careful details of our figures. We appreciate the reviewers’ enthusiasm for our in depth experiments (“ high quality in its rigorous experimental design” HALe, “Extensive set of experiments” NFbK, “Comprehensive evaluation of CoT prompting strategies across different model sizes and benchmarks.” 8tPn) and novel methodological decomposition of the prompting space (“Interesting idea of splitting the CoT prompts”, NFbK, “unique contribution by systematically exploring multi-turn code generation”, HALe, “negative results interesting”, 8tPn).

To address the reviewers’ concerns, we upload a new version of the paper, with the following updates:
- **Appendix B.4**: an extensive evaluations of the RFT model on a larger variety of benchmarks (HumanEval+, MBPP+, LiveCodeBench) which show the generalization of the finetuned model and resolve the overfitting concern of Reviewer 8tPn.
- **Appendix B.2, B.3**: we enrich all the experiments details for data collection, including dataset statistics and the configuration for deduplication, decontamination and hyperparameters for finetuning, as suggested by Reviewer 8tPn.
- **Appendix D.2**: we add a single-turn repeated sampling performance under equal code attempt to compare with different execution feedback, as suggested by Reviewer NfbK.
- **Appendix E.1**: we add ablation of retrying prompt specific to reasoning about tests failure and fixing, as suggested by Reviewer NfbK.
- **Appendix E.3**: we add a more complete ablation to RFT, including problem-solution pairs without CoT response and a SFT baseline, as suggested by Reviewer NfbK.
- **Appendix K**: we add **upper bound performance estimation** of adaptive prompt and execution feedback selection, i.e., if using different prompts per problem and the CoT prompts and the execution feedback granularity are selected by an Oracle, as suggested by all Reviewers, HALe, NfbK and 8tPn.
- Figure 2 and Figure 7: we fix the eval number for base model, as spotted by Reviewer NFbK.
- Fix all small plotting mistakes, unclear phrasing and typos in the main body.

We respond in detail to all comments in our individual response to the reviewers and will be happy to address any concerns the reviewers have during the discussion period.

---

### Meta-Review · Area_Chair_zXok · 2024-12-19

**Metareview:**

This paper performs a systematic evaluation of the interplay between chain-of-thought reasoning and multi-turn generation in LLM code generation. The question of the best strategies for code generation (especially single-turn vs. multi-turn) has received significant recent interest. They identify several high-level strategies that are effective across a range of models. There are some lingering concerns about both the clarity of the writing (especially with respect to the array of different approaches and experiments considered), as well as the novelty. I especially agree that currently, the key takeaways are somewhat obfuscated (for instance, they are nowhere to be found in the title or abstract). Thus, while the reviews lean towards acceptance, I strongly encourage the authors to improve the clarity and positioning of their paper.

**Additional Comments On Reviewer Discussion:**

There was a significant amount of discussion during the rebuttal period, and the authors addressed several of the more significant concerns raised by the reviewers.

---

### Decision · Program_Chairs · 2025-01-22

Accept (Poster)